# MegaScience: Pushing the Frontiers of Open Post-Training Datasets for Science Reasoning

## Abstract

Scientific reasoning is critical for developing AI scientists and supporting human researchers in advancing the frontiers of natural science discovery. However, the open-source community has primarily focused on mathematics and coding while neglecting the scientific domain, largely due to the absence of open, large-scale, high-quality, verifiable scientific reasoning datasets. To bridge this gap, we first present **TextbookReasoning**, an open dataset featuring truthful reference answers extracted from 12k university-level scientific textbooks, comprising 650k reasoning questions spanning 7 scientific disciplines. We further introduce **MegaScience**, a large-scale mixture of high-quality open-source datasets totaling 1.25 million instances, developed through systematic ablation studies that evaluate various data selection methodologies to identify the optimal subset for each publicly available scientific dataset. Meanwhile, we build a comprehensive evaluation system covering diverse subjects and question types across 14 benchmarks, incorporating comprehensive answer extraction strategies to ensure accurate evaluation metrics. Our experiments demonstrate that our datasets achieve superior performance and training efficiency with more concise response lengths compared to existing open-source scientific datasets. Furthermore, we train Llama3.1-8B, Qwen2.5-7B, and Qwen3 series base models on MegaScience, which outperform the corresponding official instruct models in average performance (e.g., +3.24% for Qwen3-30B-A3B). In addition, **MegaScience exhibits greater effectiveness for larger and stronger models, suggesting a scaling benefit for scientific tuning**. We release our data curation pipeline, evaluation system, datasets, and nine trained models to the community to advance scientific reasoning research.

## 1 Introduction

Large Language Models (LLMs) have evolved from knowledge retrieval systems into cognitive reasoning systems (Xia et al., 2025), representing a significant milestone toward Artificial General Intelligence (AGI) (Jaech et al., 2024; Guo et al., 2025). These reasoning models have primarily focused on mathematics and coding, as these domains provide abundant datasets, established benchmarks, and well-defined verification mechanisms (Zhou et al., 2025; Tsoukalas et al., 2024; Liu et al., 2024b; Jimenez et al., 2023). Scientific reasoning represents another critical capability that is essential for developing AI scientists and assisting human researchers in advancing the frontiers of natural science (Jumper et al., 2021). However, scientific reasoning remains significantly underdeveloped compared to mathematics and coding, particularly within the open-source community.

Despite the availability of some open-source scientific reasoning datasets, several critical challenges remain unaddressed:

(1) **Unreliable benchmark evaluation**: Many open-source scientific benchmarks adopt multiple-choice formats, which, while easy to implement, oversimplify the complexity of scientific reasoning. Consequently, post-training datasets in scientific domains often follow this format to maintain distributional consistency (e.g., Nemotron-Science (Bercovich et al., 2025)). However, our observations reveal that models trained on such data exhibit inflated performance on multiple-choice evaluations but struggle significantly with computational tasks, suggesting a disconnect between benchmark performance and true reasoning ability.

(2) **Less rigorous decontamination**: Existing decontamination techniques typically rely on n-gram overlap or embedding similarity to remove potential benchmark leakage. These methods are inherently fragile, easily circumvented by minor variations in phrasing or structure, and thus fail to ensure the integrity of benchmark evaluations. We found substantial overlap with benchmarks from most existing post-training datasets on science domains.

(3) **Low-quality reference answers**: Reference answers in many scientific datasets are either scraped from web sources (e.g., NaturalReasoning (Yuan et al., 2025)) or generated by LLMs (e.g., Nemotron-Science (Bercovich et al., 2025)). Both methods suffer from increasing unreliability—web content is now saturated with AI-generated text (Spennemann, 2025; Law et al., 2025), and LLMs themselves are prone to hallucination—making it difficult to guarantee the factual accuracy and scientific rigor of the answers.

(4) **Superficial knowledge (data) distillation**: A common practice involves distilling data from large reasoning models—such as directly prompting DeepSeek-R1 (Guo et al., 2025) to generate long chain of thoughts (CoT) (Wei et al., 2022) solutions (e.g., NaturalThoughts (Li et al., 2025) and Nemotron-Science (Bercovich et al., 2025)). While intuitive and easy to implement, it remains largely superficial. The resulting CoT data are often prone to overthinking (Chen et al., 2024b), which also brings challenges in training especially for small models and inference efficiency. Such shallow operations hinder the more principled, efficient, and generalizable knowledge transfer.

To bridge this gap, we first introduce **TEXTBOOKREASONING** (§2), an open-source university-level scientific post-training dataset with truthful reference answers, extracted from nearly 12k university-level scientific textbooks, comprising 650k reasoning questions spanning various topics, including physics, biology, chemistry, medicine, computer science, mathematics, and economics. Specifically, our data curation pipeline consists of textbook digitalization, dual QA pairs extraction, deduplication, QA pairs refinement, filtering, and LLM-based decontamination. This pipeline, fully automated through LLMs, facilitates the scalable acquisition of high-quality datasets.

To further advance open-source post-training datasets for scientific reasoning, we introduce **MEGA-SCIENCE** (§3), a large-scale mixture of high-quality open-source datasets consisting of 1.25 million instances. We first collect multiple public datasets, then conduct comprehensive ablation studies across different data selection methods to identify the optimal approach for each dataset, thereby contributing high-quality subsets. Furthermore, we annotate step-by-step solutions for public datasets.

To facilitate scientific reasoning development in the open-source community, we design and open-source an evaluation framework covering diverse subjects (e.g., biology and physics) and question types (e.g., multiple-choice questions and computational problems) across 14 benchmarks. This framework enables easy reproduction of our experimental results and fair comparison across different models by providing equitable treatment. Additionally, we design comprehensive answer extraction strategies to ensure the accuracy of final evaluation metrics.

Our supervised fine-tuning experiments (§4) demonstrate that our datasets not only enable efficient training and inference but also achieve state-of-the-art performance in the scientific domain. Finally, we train Llama3.1-8B, Qwen2.5-7B, and Qwen3 series base models on MEGASCIENCE, which outperform the official instruct models in average performance (e.g., +3.24% for Qwen3-30B-A3B), successfully advancing the frontiers of the open-source community in the science domain. We find that MEGASCIENCE exhibits greater effectiveness for larger and stronger models, suggesting a scaling benefit for scientific instruction tuning.

## 2 TEXTBOOKREASONING DATA CURATION

Current scientific datasets mainly come from web sources or LLM distillation, lacking large-scale, challenging, and diverse questions with verified answers. Textbooks, as human-curated sources, provide reliable, systematically organized knowledge with higher information density than web data, making them better for LLM knowledge learning, as shown in Phi model pretraining (Gunasekar et al., 2023; Li et al., 2023b). However, leveraging textbooks for post-training scientific reasoning remains underexplored. To fill this gap, we propose a pipeline to maximize educational value from textbooks, introducing TEXTBOOKREASONING, an open-source university-level scientific post-training dataset with verified reference answers. Derived from 12.8k textbooks, it contains 651k reasoning questions across diverse subjects. Figure 1 illustrates the data curation pipeline.

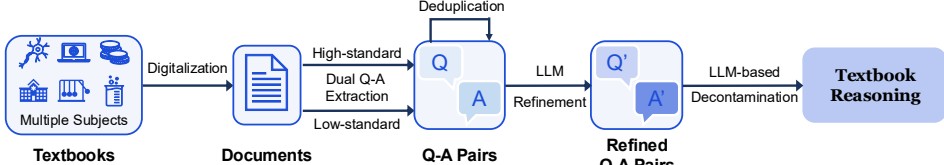

Figure 1: The pipeline of TEXTBOOKREASONING data curation.

## 2.1 TEXTBOOKS COLLECTION AND DIGITIZATION

We collected a large corpus of books by crawling PDFs online, filtering out those restricted for public access. Using Llama3.3-70B-Instruct (Grattafiori et al., 2024), we classified each book's subject and academic level, excluding materials below university level. This resulted in 12.8k academic books across seven disciplines: medicine and biology (2,305), chemistry (1,017), computer science and AI (6,057), physics (1,685), mathematics (1,578), and economics (158). Finally, we converted PDFs to machine-readable text with olmOCR (Poznanski et al., 2025).

## 2.2 DUAL Q-A PAIRS EXTRACTION AND QUESTION DEDUPLICATION

Compared to document-based question synthesis (Li et al., 2023a), Q-A pair extraction preserves more original information without introducing substantial LLM-generated content and avoids many conceptual questions such as "what is" queries. Unlike prior pipelines using a single extraction standard (Yue et al., 2024), we adopt a dual-extraction strategy with high- and low-standard criteria to comprehensively mine QAs. Textbooks were segmented into 4,096-token chunks, and each chunk was processed with Llama3.3-70B-Instruct (see F.1 for prompts). High-standard questions require multi-step reasoning and complete procedural solutions, while low-standard questions need only a reference answer. Table 15 shows extraction statistics, revealing substantial cross-discipline variation: mathematics had >60% valid chunks, whereas others had <10%. Overall, 945k pairs were extracted.

We remove redundant questions using word-level locality-sensitive min-hashing.[1] Questions with similarity above 0.6 are excluded to avoid multiple variants targeting the same reasoning tasks.

## 2.3 Q-A PAIR REFINEMENT

Many extracted questions lack the necessary context or cite documents, while their answers often omit crucial reasoning. We use DeepSeek-V3 (Liu et al., 2024a) to refine Q-A pairs with corresponding source documents for reference (see Figure 20 for the prompt), ensuring questions include all necessary contextual information and answers provide comprehensive explanations with clear reasoning processes. Then, we use Llama3.3-70B-Instruct to identify Q-A pairs missing reasoning (∼60k instances; see Figure 21 for the prompt), and then employ DeepSeek-V3 to enrich them with explanations and reformat their answers (Fan et al., 2024). We find this step to be crucial, as it leads to a 1% improvement in average. After refinement, some questions still refer to external sources or contain contradictory/missing/invalid answers. We use Llama3.3-70B-Instruct to filter out these defective Q-A pairs (∼40k; see Figure 22 for the prompt).

## 2.4 LLM-BASED QUESTION DECONTAMINATION

Incorporating benchmark questions can make evaluation unreliable (Xu et al., 2024; Sainz et al., 2024). To prevent benchmark contamination, we check overlaps between TEXTBOOKREASONING and common scientific reasoning benchmarks for LLMs, including MMLU (Hendrycks et al., 2020), GPQA (Rein et al., 2024), MMLU-Pro (Wang et al., 2024), SuperGPQA (Du et al., 2025), SciBench (Wang et al., 2023), OlympicArena (Huang et al., 2024), ChemBench (Mirza et al., 2024), CS-Bench (Song et al., 2024), MedQA (Jin et al., 2020), MedMCQA (Pal et al., 2022), Pub-MedQA (Jin et al., 2019), GSM8K (Cobbe et al., 2021), and MATH (Hendrycks et al., 2021). Simple $n$-gram overlap fails under paraphrasing or translation, so we adopt LLM-based decontamination following Toshniwal et al. (2024) and He et al. (2025): (1) for each question, we use BGE-large-en-v1.5 embeddings (Chen et al., 2024a) to retrieve the top-5 most similar benchmark examples;

---

[1]https://github.com/ChenghaoMou/text-dedup

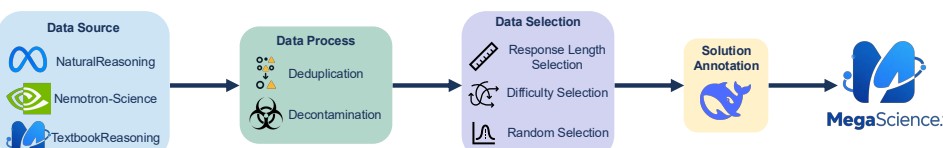

Figure 2: The overall of MEGASCIENCE data recipe.

(2) we pair each question with these examples and use Llama3.3-70B-Instruct to detect paraphrases via zero-shot prompting (see Figure 23 for the prompt). All questions identified as paraphrases are removed, resulting in the elimination of about 77k pairs. Detailed statistics for each curation step can be found in Table 16 and length statistics of each subject can be found in Table 17.

## 3 MEGASCIENCE DATA CURATION

To further advance the frontiers of open-source post-training datasets for scientific reasoning, we collect multiple public datasets and explore different data selection methods and solution annotation techniques. Ultimately, we obtain a high-quality mixed dataset, MEGASCIENCE, which consists of 1.25 million instances. An overall of the data recipe is illustrated in Figure 2.

### 3.1 SOURCING FROM PUBLIC DATASETS

We select NaturalReasoning (Yuan et al., 2025), Nemotron-Science (Bercovich et al., 2025), and our TEXTBOOKREASONING as the source datasets. We exclude SCP-116K (Lu et al., 2025) due to its inferior performance in scientific reasoning tasks.

### 3.2 QUESTION DEDUPLICATION AND DECONTAMINATION

We apply question deduplication and LLM-based question decontamination to NaturalReasoning and Nemotron-Science (details presented in §2.2 and §2.4).

### 3.3 DATA SELECTION

Since indiscriminately mixing all available data would result in reduced training efficiency, we curate high-quality subsets from each dataset and combine these refined subsets for training. We design three data selection methods:

(1) **Response Length Selection**: Following Guha et al. (2025), which demonstrated that response length selection is the optimal method for the science domain, we annotate questions with Qwen2.5-72B-Instruct and retain the questions with the longest responses.

(2) **Difficulty Selection**: To enhance reasoning, we design a two-step difficulty selection: **(a) Reference answer annotation:** For TEXTBOOKREASONING, Llama3.3-70B-Instruct generates reference answers (see Figure 24 for the prompt); NaturalReasoning uses provided references; Nemotron-Science uses DeepSeek-R1 summaries. **(b) Difficulty evaluation:** Following Tong et al. (2024), we sample 16 responses from Qwen2.5-7B-Instruct and score them with Qwen2.5-32B-Instruct (0–10) against the reference (see Figure 25 for the prompt). The average score reflects question difficulty, with lower scores indicating greater difficulty. Samples with an average score above 9 (overly easy) or below 1 (noisy) are removed.

(3) **Random Selection**: Randomly select questions.

For each dataset, we first use difficulty selection to acquire $n$ instances, and set both response length and random selection to $n$ for fair comparison. The optimal selection method per dataset is determined via supervised fine-tuning on Qwen2.5-7B (Table 1). Random selection works best for NaturalReasoning, while difficulty selection excels on Nemotron-Science. Yet, no selection method outperforms using the full TEXTBOOKREASONING, indicating few low-quality instances and supporting retention of all data in MEGASCIENCE. Step-wise statistical details are in Table 2.

### 3.4 SOLUTION ANNOTATION

Table 1: Performance comparison of data selection strategies. **General Avg.** denotes the average performance across general scientific reasoning tasks, **Specific Avg.** denotes the average performance across specific scientific reasoning tasks, and **Math Avg.** denotes the average performance across mathematical reasoning tasks (see §B.2 for details). **Bold** indicates the best results. Blue indicates the subset included in MEGASCIENCE.

| Dataset | Size (k) | General Avg. | Specific Avg. | Math Avg. | All Avg. |
|---|---|---|---|---|---|
| NaturalReasoning-DC | 1079 | 36.87 | **65.46** | **75.69** | **57.44** |
| + Response Length Selection | 436.4 | **37.70** | 63.48 | 74.76 | 56.69 |
| + Difficulty Selection | 436.4 | 36.97 | 65.07 | 75.04 | 57.17 |
| + Random Selection | 436.4 | 37.46 | 65.22 | 75.02 | 57.41 |
| Nemotron-Science-DC | 447.4 | 35.16 | 67.56 | 68.33 | 56.15 |
| + Response Length Selection | 173.3 | 34.33 | 67.43 | 71.09 | 56.39 |
| + Difficulty Selection | 173.3 | **36.71** | **68.50** | 69.67 | **57.40** |
| + Random Selection | 173.3 | 34.28 | 67.72 | 68.95 | 56.04 |
| TEXTBOOKREASONING | 651.8 | **39.58** | **65.15** | 75.93 | **58.33** |
| + Response Length Selection | 297.6 | 36.94 | 62.53 | 75.57 | 56.18 |
| + Difficulty Selection | 297.6 | 38.25 | 62.96 | 74.83 | 56.68 |
| + Random Selection | 297.6 | 37.08 | 63.46 | 73.48 | 56.18 |

For TEXTBOOKREASONING, we retain the refined solution. For NaturalReasoning, we use DeepSeek-V3 for step-by-step annotations due to low-quality responses from Llama3.3-70B-Instruct. For Nemotron-Science, DeepSeek-R1 produces overly long answers even for simple questions (Chen et al., 2024b), reducing inference efficiency; thus, we use DeepSeek-V3. To ensure quality and conciseness, responses over 4,096 tokens—often repetitive—are filtered, removing 8k instances.

Table 2: Statistics of the MEGASCIENCE dataset. **Dedup** denotes question deduplication, **DC** represents LLM-based question decontamination, and **DS** indicates data selection.

| Dataset | Raw Size | w/ Dedup | w/ DC | w/ DS |
|---|---|---|---|---|
| NaturalReasoning | 1145.8k | 1145.8k | 1079k | 436.4k |
| Nemotron-Science | 708.9k | 612k | 447.4k | 173.3k |
| TEXTBOOKREASONING | 651.8k | 651.8k | 651.8k | 651.8k |
| MEGASCIENCE | 2506.5k | 2409.6k | 2178.2k | 1261.5k |

## 4 EXPERIMENTS

We conduct supervised fine-tuning to verify the effectiveness of TEXTBOOKREASONING and MEGASCIENCE, and demonstrate the impact of each component in our data curation pipeline through comprehensive ablation studies.

### 4.1 SETUP

**Baselines** We compare our datasets with other scientific reasoning datasets:

- **SCP-116K** (Lu et al., 2025), 274K instances of questions scraped from Web and long-thought solutions generated by DeepSeek-R1.

- **NaturalReasoning** (Yuan et al., 2025), 1.1M Llama3.3-70B-instruct–synthesized general reasoning instances grounded in web sources across math, STEM, economics, social sciences, etc.

- **Nemotron-Science** (Bercovich et al., 2025), 708K diverse open-ended and multiple-choice questions combining StackOverflow extractions and synthetic MCQs; solutions generated by DeepSeek-R1 and filtered via rejection sampling.

As these baselines use n-gram overlap for decontamination, which fails under minor textual changes, we apply LLM-based benchmark decontamination (detailed in §2.4). It revealed 19K, 66K, and 164K leaked instances in SCP-116K, NaturalReasoning, and Nemotron-Science, respectively, highlighting the limitations of n-gram–based methods.

**Evaluation** Existing evaluations of scientific reasoning mainly use multiple-choice benchmarks like GPQA (Rein et al., 2024), which cover limited domains and question types. To address this, our framework includes diverse types—computational problems, judgment tasks, problem-solving questions—and domain-specific benchmarks in chemistry, CS, medicine, physics, and mathematics. For reproducibility and fair comparison, we release the full evaluation system as open source with

Table 3: The main results for scientific reasoning. All models are trained on Qwen2.5-7B. **DC** indicates LLM-based question decontamination. **Bold** indicates the best and underline indicate the second-best results.

| Subject | Benchmark | Qwen2.5-7B Instruct | SCP-116K -DC | Natural Reasoning -DC | Nemotron Science -DC | TEXTBOOK REASONING | MEGA SCIENCE |
|---------|-----------|--------------------|--------------|----------------------|---------------------|-------------------|-------------|
| General | MMLU-Pro | 56.23 | 57.75 | 52.80 | **62.87** | 55.48 | 59.16 |
| | GPQA-D | 31.31 | 29.80 | 31.31 | 29.29 | 34.34 | **36.36** |
| | SuperGPQA | 28.78 | 29.81 | 25.84 | 31.06 | 29.64 | **31.52** |
| | SciBench | 42.97 | 28.60 | 40.78 | 23.44 | 44.06 | **48.75** |
| | OlympicArena | 36.42 | 23.33 | 33.61 | 29.14 | 34.37 | **40.23** |
| Chemistry | ChemBench | 51.90 | 45.55 | 52.58 | 44.37 | 50.97 | **53.48** |
| CS | CS-Bench | 69.51 | 66.71 | 68.16 | **72.21** | 68.79 | 68.73 |
| Medicine | MedQA | 54.28 | 50.27 | 56.56 | **65.28** | 55.85 | 60.97 |
| | MedMCQA | 55.87 | 52.47 | 54.86 | 58.47 | 56.25 | 57.35 |
| | PubMedQA | 73.60 | 63.40 | 74.20 | **76.80** | 74.00 | 73.00 |
| Physics | PIQA | 86.67 | 75.30 | 86.40 | **88.25** | 85.04 | 85.80 |
| Math | GSM8K | **91.96** | 86.43 | 91.58 | 80.82 | 89.76 | 89.84 |
| | MATH | 74.90 | 74.10 | 68.90 | 66.96 | 71.44 | **76.58** |
| | MATH500 | 68.80 | 68.00 | 66.60 | 57.20 | 66.60 | **72.40** |
| Average | | 58.80 | 53.68 | 57.44 | 56.15 | 58.33 | **61.01** |

standardized templates and prompting strategies. Detailed benchmarks and setups are in Table 9. During development, we find that a key challenge is accurate **answer extraction**, as it strongly affects final accuracy. We propose a two-stage strategy: (1) detect explicit answer indicators (e.g., "The correct answer is <ANSWER>"), and (2) extract answers from textual/mathematical patterns (e.g., \boxed, \mathrm). For multiple-choice questions, we search the option content and match the corresponding option label if direct extraction of the option label fails. Extraction rules are summarized in Table 20.

The evaluation system details are provided in §B, with benchmarks organized into three categories: (1) **General Scientific Reasoning:** GPQA-Diamond (Rein et al., 2024), MMLU-Pro (Wang et al., 2024), SuperGPQA (Du et al., 2025), SciBench (Wang et al., 2023), and OlympicArena (Huang et al., 2024); (2) **Specific Scientific Reasoning:** ChemBench (Mirza et al., 2024), CS-Bench (Song et al., 2024), MedQA (Jin et al., 2020), MedMCQA (Pal et al., 2022), PubMedQA (Jin et al., 2019), and PIQA (Bisk et al., 2020); (3) **Mathematic Reasoning:** GSM8K (Cobbe et al., 2021), MATH (Hendrycks et al., 2021), and MATH500 (Lightman et al., 2023).

**Training Details**  We use LLaMA-Factory (Zheng et al., 2024) to fine-tune base models including Qwen2.5, Qwen3, and Llama3 series on our datasets and baselines. The hyperparameters are shown in Table 21. Unless otherwise specified, all experiments are conducted on Qwen2.5-7B.

## 4.2 MAIN RESULTS

**TEXTBOOKREASONING demonstrates superior performance across open-source scientific datasets**  As shown in Table 3, our TEXTBOOKREASONING surpasses other open-source datasets on most benchmarks, excelling in computational reasoning. Nemotron-Science performs better on multiple-choice tasks (e.g., MMLU-Pro, medicine tasks) due to its training data being entirely multiple-choice, causing distribution bias, but it struggles in computational tasks. In contrast, TEXTBOOKREASONING outperforms Nemotron-Science by 20.62% on SciBench and 5.23% on OlympicArena, while remaining competitive on multiple-choice benchmarks with only small gaps.

**MEGASCIENCE achieves state-of-the-art performance**  As shown in Table 3, our MEGASCIENCE achieves state-of-the-art results, ranking first on 7 of 14 benchmarks and second on 3 more, with an average 2.21% improvement over Qwen2.5-7B-Instruct. It excels across diverse scientific domains, notably achieving 48.75% on SciBench and 40.23% on OlympicArena, while also showing strong results on other domain benchmarks.

## 4.3 PUSHING THE FRONTIER IN SCIENCE DOMAIN WITH MEGASCIENCE

We demonstrate the broader effectiveness of MEGASCIENCE by training it on Qwen2.5 (Yang et al., 2025b), Qwen3 (Yang et al., 2025a), and Llama3.1 (Grattafiori et al., 2024) series base models with

Table 4: Comparison between models trained on MEGASCIENCE and official instruction-tuned models. **Bold** indicates the best. For fair comparison, Qwen3 adopts non-thinking mode due to our short CoT. The detailed results are shown in Table 18 and 19.

| Model | General Avg. | Specific Avg. | Math Avg. | All Avg. |
|---|---|---|---|---|
| **Llama3.1** | | | | |
| Llama3.1-8B-Instruct | 24.44 | **64.79** | **61.49** | 49.67 |
| Llama3.1-8B-MEGASCIENCE | **33.99** | 64.17 | 53.33 | **51.07** |
| **Qwen2.5** | | | | |
| Qwen2.5-1.5B-Instruct | **23.42** | **53.83** | **59.50** | **44.18** |
| Qwen2.5-1.5B-MEGASCIENCE | 20.77 | 50.67 | 56.23 | 41.19 |
| Qwen2.5-3B-Instruct | **32.31** | 59.38 | 67.72 | **51.50** |
| Qwen2.5-3B-MEGASCIENCE | 30.96 | **59.80** | **68.40** | 51.35 |
| Qwen2.5-7B-Instruct | 39.14 | 65.31 | 78.55 | 58.80 |
| Qwen2.5-7B-MEGASCIENCE | **43.20** | **66.55** | **79.61** | **61.01** |
| **Qwen3** | | | | |
| Qwen3-1.7B-Instruct | **32.46** | 52.14 | **73.82** | 49.76 |
| Qwen3-1.7B-MEGASCIENCE | 31.66 | **57.53** | 68.84 | **50.71** |
| Qwen3-4B-Instruct | 44.91 | 65.78 | **84.08** | 62.25 |
| Qwen3-4B-MEGASCIENCE | **45.80** | **66.83** | 82.34 | **62.64** |
| Qwen3-8B-Instruct | 50.45 | 69.53 | 84.02 | 65.82 |
| Qwen3-8B-MEGASCIENCE | **52.60** | **71.43** | **86.19** | **67.87** |
| Qwen3-14B-Instruct | 53.59 | 72.19 | 86.87 | 68.70 |
| Qwen3-14B-MEGASCIENCE | **58.07** | **74.21** | **88.54** | **71.52** |
| Qwen3-30B-A3B-Instruct | 55.66 | 74.61 | 87.55 | 70.62 |
| Qwen3-30B-A3B-MEGASCIENCE | **61.12** | **76.75** | **89.33** | **73.86** |

the same hyperparameters specified in Table 21. Our experimental results, Table 4, reveal three key findings that highlight the potential of MEGASCIENCE for advancing scientific domain capabilities.

- **Breaking performance barriers in science domain** Training with MEGASCIENCE improves performance across models and scales. Table 4 shows that Qwen2.5-7B, all Qwen3 models, and Llama3.1-8B trained on MEGASCIENCE outperform their official instruction-tuned counterparts, demonstrating MEGASCIENCE 's effectiveness in pushing the frontier in science.

- **Scaling benefits for larger and stronger models** We observe that MEGASCIENCE is more effective for larger models, indicating a scaling benefit of scientific instruction tuning. In Qwen2.5, performance is non-monotonic: Qwen2.5-1.5B-Instruct beats its MEGASCIENCE counterpart by 2.99%, the gap shrinks to 0.15% at 3B, then reverses with Qwen2.5-7B-MEGASCIENCE surpassing its baseline by 2.21%. Across generations, Qwen3 consistently shows MEGASCIENCE models outperform official instruct ones at all scales, with gaps growing with model size.

- **Mathematical reasoning requires sufficient model capacity** We find that mathematical reasoning requires sufficient model capacity: our dataset only improves performance over official instruction-tuned models when applied to stronger bases like Qwen2.5-7B and Qwen3-8B. We attribute this to the dataset's advanced difficulty, often involving undergraduate-level or higher concepts, which demands a capability threshold before models can effectively benefit.

## 4.4 ABLATION STUDY

**Impact of Core Components** To evaluate the role of core components in TEXTBOOKREASONING, we perform an ablation study (refer to Table 5). Removing the refinement component causes a sharp drop from 58.33% to 13.15%, showing its critical role in producing high-quality reasoning steps. We further conduct a human evaluation to deeply understand its quality and reliability (See §C). Removing the supplementary CoT reduces performance to 57.33%, indicating that complete solutions and step-by-step guidance enhance reasoning. Finally, removing decontamination slightly raises performance to 58.57%, confirming that our LLM-based decontamination effectively eliminates contaminated examples for more rigorous evaluation.

**Impact of Different Models for Refinement** Table 6 shows that DeepSeek-V3 consistently outperforms Llama3.3-70B-Instruct across all categories, indicating that more capable refinement models yield better downstream performance and that refinement quality correlates with model sophistication.

Table 5: The impact of each component.

| Dataset | General Avg. | Specific Avg. | Math Avg. | All Avg. |
|---|---|---|---|---|
| TEXTBOOKREASONING | 39.58 | **65.15** | 75.93 | 58.33 |
| w/o Decontamination | **39.87** | 65.12 | **76.65** | **58.57** |
| w/o Supplementary CoT | 37.63 | 64.54 | 75.73 | 57.33 |
| w/o Refinement | 4.32 | 20.37 | 13.42 | 13.15 |

Table 6: The impact of different models of refinement.

| Results | Llama3.3-70B -Instruct | DeepSeek -V3 |
|---|---|---|
| General Avg. | 34.23 | **37.63** |
| Specific Avg. | 63.84 | **64.54** |
| Math Avg. | 74.26 | **75.73** |
| All Avg. | 55.50 | **58.33** |

## 4.5 ANALYSIS

**Impact of Decontamination**  Existing datasets mostly use n-gram decontamination, which is easily bypassed by minor rephrasings (Yao et al., 2024). To overcome this, we applied LLM-based question decontamination (Toshniwal et al., 2024; He et al., 2025) to all datasets (§2.4 for details). Results in Table 7 show varied effects: three of four datasets degrade after decontamination, confirming its effectiveness in removing con-

Table 7: The impact of LLM-based question decontamination. **Bold** indicates the best results.

| Dataset | General Avg. | Specific Avg. | Math Avg. | All Avg. |
|---|---|---|---|---|
| SCP-116K | **35.76** | **60.29** | **77.93** | **55.31** |
| + Decontamination | 33.86 | 58.95 | 76.18 | 53.68 |
| NaturalReasoning | 36.60 | **65.77** | 74.08 | 57.13 |
| + Decontamination | **36.87** | 65.46 | **75.69** | **57.44** |
| Nemotron-Science | **35.79** | **67.60** | **69.30** | **56.60** |
| + Decontamination | 35.16 | 67.56 | 68.33 | 56.15 |
| TEXTBOOKREASONING | **39.87** | 65.12 | **76.65** | **58.57** |
| + Decontamination | 39.58 | **65.15** | 75.93 | 58.33 |

taminated samples. SCP-116K drops the most, indicating heavy contamination; Nemotron-Science shows modest decreases, suggesting inflated original performance. In contrast, NaturalReasoning improves, implying a lower contamination rate.

**Performance-Efficiency Trade-off Analysis**  A key challenge in reasoning model development is balancing performance and efficiency. While long CoTs are often used to boost performance, our analysis of open-source scientific reasoning datasets reveals a *counterintuitive phenomenon*.

(1) Examining training efficiency vs. performance, we compare average training response length with downstream performance of Qwen2.5-7B models. As Figure 4 shows, we observe a negative correlation: longer training responses often lead to worse performance, which we attribute to poor question quality and difficulty. This explains why naive distillation from models like DeepSeek-R1, despite long CoTs, underperforms—yielding neither efficient nor performant solutions. In contrast, our high-quality TEXTBOOKREASONING and MEGASCIENCE, with short CoTs, achieve the best trade-off, supporting both strong performance and efficient training.

(2) Examining inference efficiency vs. performance, we analyze the relationship between average response length across benchmarks and inference performance. As shown in Figure 5, models trained on short CoTs in MEGASCIENCE can dynamically generate long and detailed reasoning during inference, resulting in higher average response lengths compared to training. This adaptation enables strong generalization and substantial performance gains. Moreover, we find that Qwen3-8B-MEGASCIENCE produces shorter average responses (1080 tokens) than Qwen2.5-7B-MEGASCIENCE (1345 tokens), showing that larger models can generate more concise yet effective reasoning.

**Comparison Between Difficulty-Aware Distillation and Refinement**  We applied difficulty selection (§3.3) to TEXTBOOKREASONING, identifying 55k low-score problems as challenging. We employed DeepSeek-V3 to generate step-by-step solutions for these questions and compared them to the original refined solutions. As Table 8 shows, refinement slightly outperforms difficulty-aware distillation, likely because it uses reference documents to reduce hallucinations. Distillation, while producing longer CoT reasoning based solely on the model's knowledge, is more prone to hallucinations but shows notable gains in mathematical reasoning, highlighting the benefit of long CoT for math tasks.

Table 8: Comparison of difficulty-aware distillation and refinement approaches using DeepSeek-V3 across both datasets. **Bold** indicates the best.

| Results | Distillation | Refinement |
|---|---|---|
| General Avg. | 38.84 | **39.58** |
| Specific Avg. | **65.43** | 65.15 |
| Math Avg. | **76.39** | 75.93 |
| All Avg. | 58.28 | **58.33** |

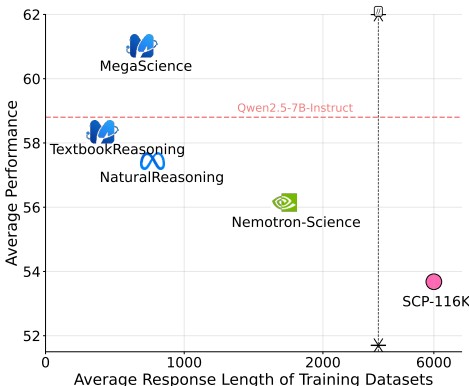

Figure 4: Trade-off between model performance and inference efficiency (average response length) on Qwen2.5-7B.

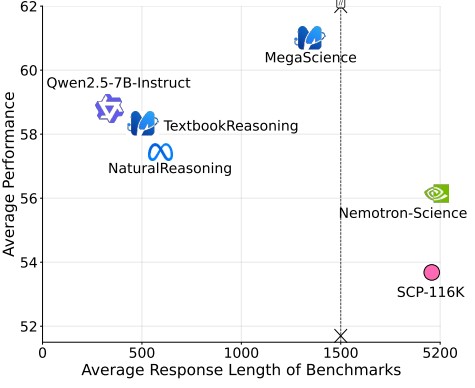

Figure 5: Trade-off between model performance and average length of responses on all benchmarks of Qwen2.5-7B.

**Question Difficulty Analysis** To estimate question difficulty, we follow Yuan et al. (2025) by using Qwen2.5-72B-Instruct to generate responses and take response length as a proxy, since longer CoTs indicate more complex questions. As Figure 3 shows, NaturalReasoning has the longest average response length (1124.7 tokens), while TEXTBOOKREASONING has a shorter average (898.5 tokens) but a broader, flatter distribution, reflecting greater variance and diversity in question difficulty. In contrast, NaturalReasoning and Nemotron-Science have more concentrated distributions, indicating more homogeneous difficulty levels.

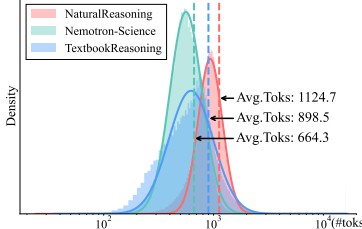

Figure 3: Response token length distributions of Qwen2.5-72B-Instruct across three datasets.

## 5 RELATED WORKS

LLMs' scientific capabilities have gained attention recently, with focus shifting from knowledge recall to reasoning due to test-time scaling (Xia et al., 2025). Current scientific reasoning datasets are built via two main approaches: (1) scraping web questions (Lu et al., 2025; Yuan et al., 2025; Ma et al., 2025; Guha et al., 2025; Li et al., 2025), with answers extracted from documents, generated by LLMs with documents, or via reasoning models like DeepSeek-R1; (2) LLMs synthesizing questions and solutions from seed data (Bercovich et al., 2025). These methods face key limitations: low-quality reference answers due to hallucination, overthinking and inefficiency from direct distillation, weak n-gram-only decontamination, and reliance on multi-choice benchmarks (e.g., MMLU (Hendrycks et al., 2020), GPQA (Rein et al., 2024)) that inadequately capture true reasoning skills like computation. To overcome these, we propose: (1) using textbooks as a reliable data source for higher-quality answers; (2) applying data selection and short CoT annotation via DeepSeek-V3 to avoid overthinking and inefficiency; (3) LLM-based benchmark decontamination to remove semantically similar data beyond n-grams; (4) designing the open-source Language Model Open Science Evaluation, covering 15 scientific benchmarks with diverse question types (multi-choice, computational, true/false, open-ended) for a more accurate assessment of reasoning abilities.

## 6 CONCLUSION

We introduce TEXTBOOKREASONING, a comprehensive open-source university-level scientific post-training dataset with 650k challenging questions and step-by-step solutions from authoritative textbooks, and MEGASCIENCE, the largest collection of high-quality open-source datasets with 1.25M instances. Through systematic experiments on data selection, we identify optimal curation strategies, providing guidelines for assembling domain-specific datasets. Supervised finetuning on Qwen-2.5, Qwen-3, and Llama3 models shows that our datasets significantly improve scientific reasoning over official instruct counterparts. We release our pipeline, datasets, evaluation system, and models to foster further advances in scientific reasoning.

ETHICS STATEMENT

During the collection and creation of our datasets, we have strictly adhered to the copyright and licensing requirements of all data sources. Specifically, we gathered open-access textbooks available on the internet without obtaining explicit consent from each individual author. We acknowledge that this approach may carry potential copyright concerns.

To mitigate and manage these risks, we have implemented the following measures: (1) Strict Filtering: We use language models to detect and filter out content from books that are strictly copyrighted or explicitly prohibited from distribution, thereby avoiding the use of such materials. (2) Research-Only Use: The datasets we create are intended solely for research purposes and are not used for any commercial activities. (3) Feedback and Removal Requests: We welcome feedback from data users and content authors at any time, including requests for removal or modification of their data, which we will promptly address.

REPRODUCIBILITY STATEMENT

We provide detailed documentation of our data curation procedures in Sections §2 and §3, covering the construction of both TEXTBOOKREASONING and MEGASCIENCE. To facilitate transparency and reproducibility, we will release the complete pipeline code. Our evaluation framework is described in Section §B, where we explicitly highlight the importance of reproducibility; the entire evaluation system will also be made publicly available. Furthermore, Table 21 reports all hyperparameters used in our training experiments to enable faithful replication of our results.

THE USE OF LARGE LANGUAGE MODELS (LLMS)

In our data construction pipeline, LLMs are employed for multiple processing tasks, including QA pair extraction, refinement, and filtering, as well as reference answer annotation and chain-of-thought annotation. For each of these steps, we explicitly report the specific LLMs and prompts used in the paper. In addition, we utilize LLMs (e.g., ChatGPT) to assist with language polishing during writing.

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

## A  DISCUSSION

**On the Relationship Between Optimal Data Mixture and Model Capability**    Our findings reveal that identifying a universally optimal post-training data mixture remains challenging across all base models. Models exhibit significant variations in capacity—whether across different architectures, parameter scales, or generational updates (e.g., Qwen2.5 vs. Qwen3). In this context, such divergence manifests as fundamentally distinct baselines in domain-specific knowledge (e.g., science). Consequently, less capable models—such as Llama series or smaller-scale Qwen2.5 instances—exhibit significant learning struggles when processing complex reasoning datasets like MEGASCIENCE without supplemental foundational data or lower-difficulty "warmup" training. These struggles manifest concretely in suboptimal responses during inference, characterized by abbreviated response length and elevated repetition rates.

**The Proxy Model Pitfall in Data Development**    When iterating on data quality or studying mixture strategies, reliance on a proxy model for validation is indispensable—yet perilous. In this work, our use of Qwen2.5-7B as a proxy tightly couples experimental outcomes and optimized data mixtures to this specific model's capabilities. While MEGASCIENCE data yields significant gains for Qwen2.5-7B, models with lower capacity struggle to replicate these results, necessitating demystification and accessibility adaptations of the data. This underscores a critical caveat: *Proxy model selection inherently biases data development, urging deliberate consideration of capability alignment and broader generalizability in future research.*

## B  MEGASCIENCE EVALUATION FRAMEWORK

We designed our evaluation framework for MEGASCIENCE and the baseline models with the following objectives: (1) **Reproducibility**: Our evaluations should be fully reproducible to ensure reliable comparisons. (2) **Comprehensive coverage**: Our evaluations should encompass diverse test domains (e.g., medicine, physics, and chemistry) and question types (e.g., multiple-choice questions and computational problems). (3) **Comparison fairness**: Our evaluation setup, including templates and prompting strategies, should provide equitable treatment across different models. (4) **Accurate answer extraction**: Our evaluation should reliably extract answers from model responses, as the answer extraction methodology significantly impacts final accuracy metrics.

Accordingly, our framework consists of four key components: an open evaluation toolkit for reproducible evaluations (§ B.1), a comprehensive suite for evaluating the scientific reasoning abilities of LLMs (§ B.2), a series of answer extraction strategies (§ B.3), and a set of recommended evaluation settings based on our experiments with various models (Table 9).

### B.1  LANGUAGE MODEL OPEN SCIENCE EVALUATION

To promote standardized and reproducible evaluations, we will open-source the codebase used to conduct all evaluations in this work. Our open science evaluation system offers the following features:

- Support for both conversation models and base models;
- Easy integration of new benchmarks and configurations (e.g., prompting and few-shot settings);
- Scalable evaluation of multiple models, benchmarks, and tasks in a single run with multi-node and multi-GPU parallelization;
- Comprehensive instance-level output data enabling fine-grained analysis of model predictions.

### B.2  MEGASCIENCE EVALUATION SUITE

To comprehensively evaluate scientific abilities, our evaluation framework encompasses both general science knowledge and specialized subject areas across multiple question formats. Below, we introduce our category and the included benchmarks.

- **General Scientific Reasoning:** GPQA-Diamond (Rein et al., 2024), MMLU-Pro (Wang et al., 2024), SuperGPQA (Du et al., 2025), SciBench (Wang et al., 2023), and OlympicArena (Huang et al., 2024).

Table 9: The MEGASCIENCE evaluation settings. **CoT** denotes evaluations conducted with chain-of-thought prompting. **Unit** indicates that the answer requires unit assignment. **EM (unit)** represents exact match accuracy for both the numerical answer and its corresponding unit.

| Category | Benchmark | Question Type | CoT | Unit | Metric |
|---|---|---|---|---|---|
| General Reasoning | GPQA-Diamond | Multi-Choice | ✓ | ✗ | EM |
| | MMLU-Pro | Multi-Choice | ✓ | ✗ | EM |
| | SuperGPQA | Multi-Choice | ✓ | ✗ | EM |
| | SciBench | Computational Problems | ✓ | ✓ | EM (unit) |
| | OlympicArena | Computational Problems | ✓ | ✓ | EM (unit) |
| Chemistry | ChemBench | Multi-Choice & Problem-Solving | ✓ | ✗ | EM |
| Computer Science | CS-Bench | Multi-Choice & True/False | ✓ | ✗ | EM |
| Medicine | MedQA | Multi-Choice | ✓ | ✗ | EM |
| | MedMCQA | Multi-Choice | ✓ | ✗ | EM |
| | PubMedQA | Multi-Choice | ✓ | ✗ | EM |
| Physics | PIQA | Multi-Choice | ✓ | ✗ | EM |
| Math | GSM8K | Computational Problems | ✓ | ✗ | EM |
| | MATH | Computational Problems | ✓ | ✗ | EM |
| | MATH500 | Computational Problems | ✓ | ✗ | EM |

- **Specific Scientific Reasoning:** ChemBench (Mirza et al., 2024), CS-Bench (Song et al., 2024), MedQA (Jin et al., 2020), MedMCQA (Pal et al., 2022), PubMedQA (Jin et al., 2019), and PIQA (Bisk et al., 2020).
- **Mathematic Reasoning:** GSM8K (Cobbe et al., 2021), MATH (Hendrycks et al., 2021), and MATH500 (Lightman et al., 2023).

### B.3 ANSWER EXTRACTION STRATEGY

Answer extraction is critically important for evaluation, as extraction accuracy can substantially impact overall results. Many scientific evaluations simply extract content within `\boxed{}`, often omitting responses that lack this formatting and incorrectly attributing such formatting errors to reduced overall accuracy. To enhance extraction precision, we develop a comprehensive set of rule-based methods tailored to extract answers across diverse question types. Our answer extraction method operates through a two-stage process: (1) identifying answer indicator phrases that signal the presence of a final answer, and (2) extracting the answer content from various formatting patterns. For answer indicators, we recognize patterns such as `The final answer to this question is <ANSWER>` and `The correct answer is <ANSWER>`. For answer formats, we handle multiple mathematical and textual formatting styles including `\boxed{}`, `\mathrm{}`, and `\mathbf{}`. The complete set of extraction rules is provided in Table 20. Moreover, for multiple-choice questions, we search the option content and match the corresponding option label if direct extraction of the option label fails.

## C TRAINING DETAILS

The training details is shown in Table 21.

## D HUMAN EVALUATION

The goal of applying DeepSeek-V3 in the refinement and solution-annotation stages is to exploit its generative and reformulation capabilities—such as improving clarity, structure, and step-by-step reasoning—rather than to rely on its creative knowledge generation. This design choice aims to reduce hallucination risks by constraining the model to transformation-oriented tasks. Nevertheless,

Table 10: Pairwise comparison of refined vs. original questions.

|  | Refined Better | Tie | Refined Worse |
|---|---|---|---|
| Count | 70 | 29 | 1 |

Table 11: Pairwise comparison of refined vs. original answers.

|  | Refined Better | Tie | Refined Worse |
|---|---|---|---|
| Count | 95 | 3 | 2 |

hallucinations and other substantive errors are still difficult to avoid entirely in practice. Therefore, it is crucial to quantitatively assess the reliability of the refined content and provide future users with a clear understanding of the strengths and limitations of this refinement process.

To this end, we conducted a human evaluation on 100 randomly sampled instances from TEXT-BOOKREASONING. Each instance contains an original question–answer pair and its corresponding refined version. The evaluation focuses on two dimensions: (1) pairwise comparison between original and refined content, and (2) error rate and error taxonomy.

### D.1 PAIRWISE COMPARISON OF QUESTION AND ANSWER QUALITY

Annotators performed a pairwise comparison between the original and refined versions of both questions and answers. Each comparison was labeled as:

- **Refined Better**: the refined version is clearer, more complete, more self-contained, or better structured;
- **Tie**: the two versions are essentially equivalent in clarity, correctness, or usefulness;
- **Refined Worse**: the refined version introduces issues or becomes less clear or less faithful than the original.

Results for question comparison are shown in Table 10. The refined question is preferred in 70% of the cases, tied in 29%, and worse in 1%. The majority of ties occur when the original question is already self-contained, making further refinement unnecessary or resulting only in minor cosmetic edits.

Results for answer comparison are presented in Table 11. The refined answer is preferred in 95% of cases, tied in 3%, and worse in 2%. These results indicate that DeepSeek-V3 is particularly effective at improving solution trajectories, often producing more structured, complete, and pedagogically useful responses.

Overall, the pairwise evaluation demonstrates that DeepSeek-V3 provides consistent quality improvements. This suggests that the model's generative and reformulation capabilities are effective in enhancing clarity, reasoning structure, and answer completeness, even though occasional errors still occur.

### D.2 ERROR RATE AND ERROR TAXONOMY

To directly assess potential hallucinations or other substantive issues, annotators additionally mark whether each refined question or answer contains any error that affects correctness or usability. Overall error counts are summarized in Table 12: 7% of refined questions and 6% of refined answers are judged to contain non-trivial issues.

Table 12: Error rates of refined questions and answers.

| Item Type | # with Error |
|---|---|
| Refined Question | 7 |
| Refined Answer | 6 |

We further categorize these errors in Tables 13 and 14. The most common issues include undesirable external references, leftover refinement metadata, missing elements (e.g., options), or minor

Table 13: Error types identified in refined questions.

| Error Type | Count |
|---|---|
| External reference (e.g., "see Formula 1.1") | 3 |
| Invalid/meta information | 1 |
| Missing options in multiple-choice question | 1 |
| Hallucinated factual content | 1 |
| Missing necessary information | 1 |

Table 14: Error types identified in refined answers.

| Error Type | Count |
|---|---|
| Invalid/meta information | 2 |
| External reference (e.g., "see Formula 1.1") | 1 |
| Redundant option included | 1 |
| Hallucinated factual content | 1 |
| Insufficient chain-of-thought | 1 |

hallucinations. Errors in refined answers occasionally stem from insufficient reasoning steps or redundant content.

The human evaluation indicates that while the refinement process is not error-free, the overall reliability is high: the majority of refined questions and answers are strictly better than the originals, and the error rate remains low ( 5–7%). These results provide a concrete and quantitative characterization of the refinement stage and offer guidance for future users regarding its strengths and limitations.

# E  DETAILED RESULTS

The detailed statistics of TEXTBOOKREASONING are presented in Table 15, Table 16 and Table 17. The detailed results of MEGASCIENCE are shown in Table 18 and 19.

# F  PROMPTS

## F.1  PROMPTS FOR Q-A PAIRS EXTRACTION

The prompts used for Q-A pair extraction across seven domains (biology, chemistry, computer science, economics, mathematics, medicine, and physics) are presented in Figure 6–19.

## F.2  PROMPTS FOR Q-A PAIRS REFINEMENT

The prompt used for Q-A pair refinement is shown in Figure 20, the prompt for identifying answers that lack chain-of-thought reasoning is shown in Figure 21, and the prompt for filtering defective Q-A pairs is shown in Figure 22.

## F.3  PROMPTS FOR QUESTION DECONTAMINATION

The prompt used for LLM-based question decontamination is shown in Figure 23.

## F.4  PROMPTS FOR DIFFICULTY SELECTION

The prompt used for annotating reference answers is shown in Figure 24, and the prompt used for evaluating student answers is shown in Figure 25.

Table 15: Q-A Extraction Statistics

| Subject | # Books | # Chunks | # Valid Chunks | # Extracted Pairs (High) | # Extracted Pairs (Low) |
|---|---|---|---|---|---|
| Biology | 2,305 | 119,581 | 6,929 | 1,394 | 102,926 |
| Chemistry | 1,017 | 49,847 | 5,490 | 1,979 | 70,756 |
| Computer Science | 6,057 | 116,380 | 5,521 | 5,890 | 16,322 |
| Economics | 158 | 8,071 | 329 | 94 | 1,851 |
| Mathematics | 1578 | 56,952 | 35,876 | 6,376 | 553,786 |
| Medicine | 2,305 | 119,581 | 9,797 | 4,919 | 120,296 |
| Physics | 1,685 | 75,722 | 8,606 | 4,831 | 54,263 |
| Total | 12,800 | 546,134 | 72,548 | 25,483 | 920,200 |

Table 16: The numerical changes in statistics during TEXTBOOKREASONING curation.

| Actions | Biology | Chemistry | CS | Economics | Mathematics | Medicine | Physics | Total |
|---|---|---|---|---|---|---|---|---|
| Q-A Pairs | 104,320 | 72,735 | 22,212 | 1,945 | 560,162 | 125,215 | 59,094 | 945,683 |
| + Deduplication | 71,693 | 39,984 | 19,433 | 1,790 | 472,740 | 111,930 | 50,323 | 767,893 |
| + Filtering | 70,102 | 37,890 | 18,843 | 1,725 | 444,126 | 109,192 | 46,889 | 728,767 |
| + Decontamination | 52,850 | 32,157 | 17,742 | 1,296 | 424,714 | 81,638 | 41,443 | 651,840 |

Table 17: Length statistics of TEXTBOOKREASONING.

| Subject | # Pairs | Avg. Question Tokens | Avg. Answer Tokens |
|---|---|---|---|
| Biology | 52,850 | 27.17 | 295.58 |
| Chemistry | 32,157 | 67.43 | 371.32 |
| Computer Science | 17,742 | 79.54 | 393.34 |
| Economics | 1,296 | 78.10 | 301.34 |
| Mathematics | 424,714 | 70.83 | 442.95 |
| Medicine | 81,638 | 34.45 | 286.59 |
| Physics | 41,443 | 86.30 | 496.54 |
| Total | 651,840 | 63.80 | 409.66 |

Table 18: The detailed results of Llama3.1 and Qwen2.5 models trained on MEGASCIENCE and official instruction-tuned models. **Bold** indicates the best.

| Benchmark | Llama3.1 8B Instruct | Llama3.1 8B MEGASCIENCE | Qwen2.5 1.5B Instruct | Qwen2.5 1.5B MEGASCIENCE | Qwen2.5 3B Instruct | Qwen2.5 3B MEGASCIENCE | Qwen2.5 7B Instruct | Qwen2.5 7B MEGASCIENCE |
|---|---|---|---|---|---|---|---|---|
| MMLU-Pro | 45.15 | **50.03** | 30.47 | **34.79** | **45.20** | 44.91 | 56.23 | **59.16** |
| GPQA-D | 24.24 | **33.33** | **30.30** | 15.15 | **32.32** | 24.75 | 31.31 | **36.36** |
| SuperGPQA | 19.72 | **25.56** | **18.90** | 17.81 | **23.42** | 22.47 | 28.78 | **31.52** |
| SciBench | 10.78 | **34.06** | 17.81 | **18.75** | 33.12 | **36.09** | 42.97 | **48.75** |
| OlympicArena | 22.31 | **26.98** | **19.62** | 17.36 | **27.46** | 26.60 | 36.42 | **40.23** |
| ChemBench | 49.57 | **50.39** | **42.03** | 41.99 | 46.52 | **47.63** | 51.90 | **53.48** |
| CS-Bench | 57.87 | **59.62** | **56.91** | 54.61 | **64.90** | 62.82 | **69.51** | 68.73 |
| MedQA | **67.01** | 60.49 | 37.71 | **39.36** | **46.82** | 45.33 | 54.28 | **60.97** |
| MedMCQA | **57.92** | 54.08 | 41.31 | **43.13** | 48.36 | **50.51** | 55.87 | **57.35** |
| PubMedQA | **78.80** | 76.80 | **68.80** | 68.20 | 67.20 | **71.20** | **73.60** | 73.00 |
| PIQA | 77.58 | **83.62** | **76.22** | 56.75 | **82.48** | 81.34 | **86.67** | 85.8 |
| GSM8K | **83.40** | 72.10 | **73.84** | 72.86 | 80.67 | **83.02** | **91.96** | 89.84 |
| MATH | **50.48** | 46.90 | **54.66** | 49.24 | **65.68** | 62.18 | 74.90 | **76.58** |
| MATH500 | **50.60** | 41.00 | **50.00** | 46.60 | 56.80 | **60.00** | 68.80 | **72.40** |
| Average | 49.67 | **51.07** | **44.18** | 41.19 | **51.50** | 51.35 | 58.80 | **61.01** |

Table 19: The detailed results of Qwen3 series models trained on MEGASCIENCE and official instruction-tuned models. **Bold** indicates the best. For fair comparison, Qwen3 adopts non-thinking mode due to our short CoT.

| Benchmark | Qwen3 1.7B Instruct | Qwen3 1.7B MEGASCIENCE | Qwen3 4B Instruct | Qwen3 4B MEGASCIENCE | Qwen3 8B Instruct | Qwen3 8B MEGASCIENCE | Qwen3 14B Instruct | Qwen3 14B MEGASCIENCE | Qwen3 30B-A3B Instruct | Qwen3 30B-A3B MEGASCIENCE |
|---|---|---|---|---|---|---|---|---|---|---|
| MMLU-Pro | 40.87 | **43.94** | 59.42 | **60.81** | 64.89 | **66.81** | 68.61 | **71.60** | 71.78 | **73.06** |
| GPQA-D | **33.33** | 23.23 | **37.37** | 34.85 | **47.47** | 46.46 | 49.49 | **50.51** | 52.02 | **57.58** |
| SuperGPQA | **22.86** | 22.27 | 31.42 | **33.08** | 35.70 | **38.84** | 39.87 | **44.35** | 42.06 | **46.86** |
| SciBench | 33.05 | **41.09** | 51.88 | **55.00** | 56.41 | **61.25** | 58.44 | **68.13** | 59.53 | **69.22** |
| OlympicArena | **32.18** | 27.77 | 44.44 | **45.25** | 47.79 | **49.65** | 51.55 | **55.76** | 52.89 | **58.86** |
| ChemBench | 44.33 | **46.63** | 54.19 | 54.12 | 54.38 | **56.78** | 58.07 | **58.71** | 59.97 | **61.65** |
| CS-Bench | 51.52 | **60.86** | 70.92 | 70.59 | 74.69 | **76.43** | 78.18 | **79.92** | 79.08 | **81.33** |
| MedQA | 39.75 | **43.05** | 57.34 | **58.84** | 65.99 | **66.06** | 70.38 | **71.56** | 76.04 | **78.16** |
| MedMCQA | 42.31 | **47.62** | 54.79 | **58.28** | 61.18 | **63.30** | 64.79 | **66.79** | 67.68 | **69.27** |
| PubMedQA | 69.60 | **71.40** | 73.60 | **76.80** | 74.20 | **77.80** | 73.00 | **78.20** | 74.20 | **78.40** |
| PIQA | 65.34 | **75.63** | **83.84** | 82.37 | 86.72 | **88.19** | 88.74 | **90.10** | 90.70 | **91.68** |
| GSM8K | 82.03 | **82.41** | **91.74** | 91.58 | 91.89 | **93.48** | 93.86 | **94.77** | 94.62 | **94.69** |
| MATH | **73.22** | 63.90 | **83.50** | 81.44 | 83.98 | **85.30** | 86.76 | **88.24** | 87.24 | **89.90** |
| MATH500 | **66.20** | 60.20 | **77.00** | 74.00 | 76.20 | **79.80** | 80.00 | **82.60** | 80.80 | **83.40** |
| Average | 49.76 | **50.71** | 62.25 | **62.64** | 65.82 | **67.87** | 68.70 | **71.52** | 70.62 | **73.86** |

```
Below is a biology document extract. Assess whether it contains a
biology question-and-answer pair that requires reasoning:

- If the document extract does not contain a biology
question-and-answer pair that involves reasoning, return the
explicit symbol `[NO QA]`.
- If the document only contains simple factual or conceptual
questions without deeper reasoning, return `[NO QA]`.
- If a biology reasoning question-and-answer pair is found, extract
it in the following format:
    Question: <question text with complete problem statement and
    all necessary biological information>
    Answer: <complete solution with all necessary reasoning steps,
    processes, and explanations included> (only if an answer is
    provided, otherwise do not generate this line)
- The extracted pair must:
  1. Require logical or scientific reasoning beyond simple recall
  2. Be self-contained and biologically precise
  3. Include all necessary context for independent solving
  4. May involve mechanisms, pathways, evolutionary principles,
  genetic analysis, experimental design, or systems-level
  understanding

Do NOT extract simple definitional questions or basic concept
recall questions.

#### The extract:
`<DOCUMENT>`

Now process the extract and return the result.
```

Figure 6: High-standard prompt for extracting Q-A pairs of biology.

```
Below is a book document extract.

# Extraction Task
Extract complete, independently solvable biology questions and
answers from the document while following these guidelines:

## For Questions:
- Extract any explicit biology questions with their associated
answers
- For implicit biology concepts, mechanisms, processes, or
principles presented as statements, convert them to well-formed
questions ONLY if they can stand alone
- Ensure each extracted question contains ALL necessary information
to be solved independently without requiring additional context
- Include any relevant biological diagrams, pathways, or figures
mentioned (describe them if not visible)
- Extract multiple questions separately if they exist
- If no biological content can be meaningfully extracted as a
question, return `[NO QA]`

## For Answers:
- Include the answer provided in the extract
- Answers should capture the essential explanation of the
biological concept
- If the source material contains a description of a mechanism or
pathway, include this in the answer
- For biological processes, the answer should explain the function,
steps, or significance as presented in the text

## Format:
Format each question-answer pair as:
Question: [Complete biology question with all context needed to
understand]
Answer: [Corresponding answer from the text]

The extract is as follows:
`<DOCUMENT>`

Now process the extract and return the result.
```

Figure 7: Low-standard prompt for extracting Q-A pairs of biology.

```
Below is a chemistry document extract. Assess whether it contains a
chemistry question-and-answer pair requiring significant reasoning:

- If the document extract does not contain a chemistry reasoning
question-and-answer pair, return the explicit symbol`[NO QA]`.
- If a chemistry question-and-answer pair requiring reasoning is
found, extract it in the following format:
    Question: <question text with complete problem statement and
    all necessary chemical information>
    Answer: <complete solution with all necessary steps, equations,
    calculations, and reasoning included> (only if an answer is
    provided, otherwise do not generate this line)
- The extracted pair must:
  1. Require chemical reasoning or multi-step problem-solving (not
  simple definition or concept recall)
  2. Be self-contained and chemically precise, allowing independent
  solving without additional context
  3. Involve topics such as: reaction mechanisms, thermodynamics,
  equilibrium calculations, acid-base chemistry, electrochemistry,
  kinetics, spectroscopic analysis, or other areas requiring
  deductive reasoning
- Do NOT extract simple definitional questions, basic concept
recalls, or single-step calculations.

#### The extract:
`<DOCUMENT>`

Now process the extract and return the result.
```

Figure 8: High-standard prompt for extracting Q-A pairs of chemistry.

Table 20: Answer Extraction Patterns

|  |  |
|---|---|
| **Answer Indicators** | The final answer to this question is <ANSWER>
The correct answer is <ANSWER>
The best option is <ANSWER>
The answer is <ANSWER>
Answer: <ANSWER>
Answer should be: <ANSWER>
Answer must be <ANSWER>
Answer is probably <ANSWER>
<ANSWER> is correct
<ANSWER> seems correct
<ANSWER> is the right answer
Answer is <ANSWER>
... |
| **Answer Formats** | \boxed{}
\mathrm{}
\mathbf{}
\text{}
()
[] |

```
Below is a book document extract.

# Extraction Task
Extract complete, independently solvable chemistry questions and
answers from the document while following these guidelines:

## For Questions/Problems:
- Extract any explicit chemistry questions with their answers
- Extract ONLY questions that are completely self-contained and can
be solved independently
- For implicit problems (chemical principles, reactions, or
concepts presented as statements), convert them to well-formed
questions ONLY if they can stand alone
- Ensure each extracted problem contains ALL necessary information
to be solved independently
- Include any relevant diagrams, figures, or charts mentioned
(describe them if not visible)
- Extract multiple problems separately if they exist
- If no mathematical content can be extracted, return `[NO QA]`

## For Answers:
- Include the complete answer if provided in the extract
- Answers should contain the main solution or explanation
- If a detailed step-by-step solution is available, include it
- For reaction mechanisms, include all steps and intermediates

## Format:
Format each question-answer pair as:
Question: [Complete chemistry question with all context needed to
solve]
Answer: [Complete answer]

The extract is as follows:
`<DOCUMENT>`

Now process the extract and return the result.
```

Figure 9: Low-standard prompt for extracting Q-A pairs of chemistry.

Table 21: Hyperparameters of supervised finetuning.

|  | LR | LR Schedule | Batch Size | Max Length | Warm Up Ratio | Epochs |
|---|---|---|---|---|---|---|
| SCP-116K | 5e-6 | Cosine | 128 | 16,384 | 0.05 | 3 |
| NaturalReasoning | 5e-6 | Cosine | 512 | 4,096 | 0.05 | 3 |
| Nemotron-Science | 5e-6 | Cosine | 128 | 16,384 | 0.05 | 3 |
| TEXTBOOKREASONING | 5e-6 | Cosine | 512 | 4,096 | 0.05 | 3 |
| MEGASCIENCE | 5e-6 | Cosine | 512 | 4,096 | 0.05 | 3 |

```
Below is a document extract. Assess whether it contains a computer
science or artificial intelligence question-and-answer pair that
requires significant reasoning:

- If the document extract does not contain a computer science or
artificial intelligence question-and-answer pair requiring
reasoning, return the explicit symbol `[NO QA]`.
- If the extract contains only simple definitional or conceptual
questions without reasoning, return the explicit symbol `[NO QA]`.
- If a reasoning-based computer science or artificial intelligence
question-and-answer pair is found, extract it in the following
format:
    Question: <complete problem statement including all necessary
    information, constraints, and requirements>
    Answer: <complete solution with all necessary reasoning steps,
    algorithms, code snippets, or formal proofs> (only if an answer
    is provided, otherwise do not generate this line)
- The extracted pair must be self-contained and technically precise,
allowing independent solving without additional context.
- Prioritize questions that involve algorithm design, computational
complexity analysis, system architecture decisions, AI model
reasoning, optimization problems, or formal proofs.
- Do not extract simple factual questions about technology history,
basic definitions, or conceptual explanations that don't require
problem-solving.

#### The extract:
`<DOCUMENT>`

Now process the extract and return the result.
```

Figure 10: High-standard prompt for extracting Q-A pairs of computer science and artificial intelligence.

```
Below is a book document extract.

# Extraction Task
Extract complete, independently solvable computer science and
artificial intelligence questions and answers from the document
while following these guidelines:

## For Questions/Problems:
- Extract any explicit computer science or AI questions with their
provided answers
- For implicit problems (algorithms, data structures, programming
concepts, AI theories, computational theorems, or technical
definitions presented as statements), convert them to well-formed
questions ONLY if they can stand alone as complete problems
- Ensure each extracted problem contains ALL necessary information
to be solved independently without requiring additional context
- Include all context, requirements, constraints, and examples
needed to understand the problem
- For computational problems, make sure the question includes all
necessary inputs, expected outputs, and constraints
- Extract multiple problems separately if they exist
- If no computer science or AI content can be extracted as complete
questions, return `[NO QA]`

## For Answers:
- Include the complete answer as provided in the extract
- Answers should contain the main solution or explanation
- If available, include:
  * Code implementations
  * Time/space complexity analysis
  * Step-by-step explanations
  * Proofs for computational theorems
  * Practical implementation details for AI concepts

## Format:
Format each question-answer pair as:
Question: [Complete computer science/AI question with all context
needed to solve]
Answer: [Complete answer]

The extract is as follows:
`<DOCUMENT>`

Now process the extract and return the result.
```

Figure 11: Low-standard prompt for extracting Q-A pairs of computer science and artificial intelligence.

```
Below is a document extract on economics. Assess whether it
contains a challenging economics question-and-answer pair that
requires reasoning:

- If the document extract does not contain a challenging economics
question-and-answer pair requiring reasoning, return the explicit
symbol `[NO QA]`.
- If the document extract contains only simple conceptual
definitions or basic knowledge, return `[NO QA]`.
- If a challenging economics question-and-answer pair requiring
reasoning is found, extract it in the following format:
    Question: <question text with complete problem statement and
    all necessary economic information>
    Answer: <complete solution with all necessary reasoning steps,
    economic analysis, and calculations included> (only if an
    answer is provided, otherwise do not generate this line)
- The extracted pair must be self-contained and economically
precise, allowing independent solving without additional context.

#### The extract:
`<DOCUMENT>`

Now process the extract and return the result.
```

Figure 12: High-standard prompt for extracting Q-A pairs of economics.

Below is a book document extract.

# Extraction Task
Extract complete, independently solvable economics questions and answers from the document while following these guidelines:

## For Questions/Problems:
- Extract any explicit economics questions with their answers
- Extract ONLY questions that are completely self-contained and can be solved independently
- For implicit problems (economic principles, models, theorems, or concepts presented as statements), convert them to well-formed questions ONLY if they can stand alone
- Ensure each extracted problem contains ALL necessary information to be solved independently
- For computational problems (supply/demand analysis, equilibrium pricing, cost-benefit calculations, elasticity, utility maximization, game theory payoffs, etc.), include all required data and parameters
- Include any relevant diagrams, figures, graphs, or tables mentioned (describe them if not visible)
- Extract multiple problems separately if they exist
- If no economics content can be extracted, return `[NO QA]`

## For Answers:
- Include the complete answer if provided in the extract
- Answers should contain the main solution or explanation
- If a detailed step-by-step solution is available, include it
- For model derivations or theoretical proofs, include all steps and reasoning

## Format:
Format each question-answer pair as:
Question: [Complete economics question with all context needed to solve]
Answer: [Complete answer]

The extract is as follows:
`<DOCUMENT>`

Now process the extract and return the result.

Figure 13: Low-standard prompt for extracting Q-A pairs of economics.

```
Below is a math document extract. Assess whether it contains a
mathematical question-and-answer pair:

- If the document extract does not contain a mathematical
question-and-answer pair, return the explicit symbol`[NO QA]`.
- If a mathematical question-and-answer pair is found, extract it
in the following format:
    Question: <question text with complete problem statement and
    all necessary mathematical information>
    Answer: <complete solution with all necessary steps and
    calculations included> (only if an answer is provided,
    otherwise do not generate this line)
- The extracted pair must be self-contained and mathematically
precise, allowing independent solving without additional context.

#### The extract:
`<DOCUMENT>`

Now process the extract and return the result.
```

Figure 14: High-standard prompt for extracting Q-A pairs of math.

```
Below is a book document extract.

# Extraction Task
Extract complete, independently solvable mathematical content
following these guidelines:

## For Questions/Problems:
- Extract any explicit mathematical questions with their answers
- Convert mathematical theorems, propositions, definitions, or
problems without explicit questions into well-formed questions
- Ensure each extracted problem contains ALL necessary information
to be solved independently
- Include any relevant diagrams, figures, or charts mentioned
(describe them if not visible)
- Extract multiple problems separately if they exist
- If no mathematical content can be extracted, return `[NO QA]`

## For Answers:
- Include the provided solution, proof, or explanation when
available
- Brief answers are acceptable if that's all the source provides
- For theorems/propositions, the question should ask to prove the
statement

## Format:
Question: <Complete mathematical problem with all context needed to
solve>
Answer: <Solution as provided in the extract>

The extract is as follows:
`<DOCUMENT>`

Now process the extract and return the result.
```

Figure 15: Low-standard prompt for extracting Q-A pairs of math.

```
Below is a medical document extract. Assess whether it contains a
medical question-and-answer pair that requires clinical reasoning:

- If the document extract does not contain a medical reasoning
question-and-answer pair, return the explicit symbol `[NO QA]`.
- If a medical reasoning question-and-answer pair is found, extract
it in the following format:
    Question: <question text with complete clinical scenario and
    all necessary patient information>
    Answer: <complete solution with diagnostic reasoning,
    differential diagnoses, management plan, and treatment
    rationale> (only if an answer is provided, otherwise do not
    generate this line)
- Only extract complex questions requiring clinical reasoning,
diagnosis, or treatment planning. Do not extract simple factual or
concept-based questions.
- The extracted pair must be self-contained and medically precise,
allowing independent assessment without additional context.
- Focus on cases requiring differential diagnosis, interpretation
of lab results, management decisions, or therapeutic reasoning.

#### The extract:
`<DOCUMENT>`

Now process the extract and return the result.
```

Figure 16: High-standard prompt for extracting Q-A pairs of medicine.

```
Below is a medical document extract.

# Extraction Task
Extract complete, independently solvable medical questions and
answers from the document while following these guidelines:

## For Questions:
- Extract any explicit medical questions with their corresponding
answers
- For implicit medical cases, conditions, diagnoses, or treatment
protocols, convert them into well-formed questions ONLY if they can
stand alone
- Ensure each extracted question contains ALL necessary clinical
information to be understood and answered independently
- Include any relevant patient data, symptoms, test results, or
clinical observations needed to fully understand the case
- Extract multiple questions separately if they exist
- If no medical question content can be extracted, return `[NO QA]`

## For Answers:
- Include the complete answer if provided in the extract
- Focus on capturing the main diagnosis, treatment plan, or
clinical explanation
- Answers should be self-contained but don't need to be exhaustive
- Include key points from any detailed explanations or management
plans provided

## Format:
Format each question-answer pair as:
Question: [Complete medical question with all context needed to
understand the case]
Answer: [Complete answer with diagnosis, treatment, or explanation]

The extract is as follows:
`<DOCUMENT>`

Now process the extract and return the result.
```

Figure 17: Low-standard prompt for extracting Q-A pairs of medicine.

```
Below is a physics document extract. Assess whether it contains a
physics question-and-answer pair that requires significant
reasoning:

- If the document extract does not contain a physics
question-and-answer pair requiring substantial reasoning, return
the explicit symbol `[NO QA]`.
- If the document extract contains only simple conceptual
definitions or basic physics facts without reasoning steps, return
`[NO QA]`.
- If a physics question-and-answer pair requiring reasoning is
found, extract it in the following format:
    Question: <question text with complete problem statement and
    all necessary physics information, including any relevant
    diagrams, equations, or quantities>
    Answer: <complete solution with all necessary reasoning steps,
    calculations, and physical principles applied> (only if an
    answer is provided, otherwise do not generate this line)
- The extracted pair must be self-contained and physically precise,
allowing independent solving without additional context.

#### The extract:
`<DOCUMENT>`

Now process the extract and return the result.
```

Figure 18: High-standard prompt for extracting Q-A pairs of physics.

```
Below is a book document extract.

# Extraction Task
Extract complete, independently solvable physics questions and
answers from the document while following these guidelines:

## For Questions/Problems:
- Extract any explicit physics questions with their answers
- Extract ONLY questions that are completely self-contained and can
be solved independently
- For implicit problems (physics principles, laws, theorems, or
concepts presented as statements), convert them to well-formed
questions ONLY if they can stand alone
- Ensure each extracted problem contains ALL necessary information
to be solved independently
- Include any relevant diagrams, figures, or charts mentioned
(describe them if not visible)
- Extract multiple problems separately if they exist
- If no physics content can be extracted, return `[NO QA]`

## For Answers:
- Include the complete answer if provided in the extract
- Answers should contain the key solution or explanation with
minimal detail
- For calculation problems, include the relevant formulas, key
steps, and final answer with units
- For derivations, include the main steps of the derivation

## Format:
Format each question-answer pair as:
Question: [Complete physics question with all context needed to
solve]
Answer: [Complete answer]

The extract is as follows:
`<DOCUMENT>`

Now process the extract and return the result.
```

Figure 19: Low-standard prompt for extracting Q-A pairs of physics.

```
Below is a question-and-answer pair and a reference document. Your
task is to refine the question to make it clear and self-contained,
then verify and refine the answer to ensure it's correct and
well-explained.

For the question:
- Ensure it contains sufficient information to be understood
independently
- Add necessary context from the reference document if the question
is unclear
- Maintain the original question's intent

For the answer:
- Verify correctness against the reference document
- If incorrect, provide the correct answer based on the document
- If the answer lacks explanation, add necessary intermediate
reasoning process leading to the given answer as a teacher would
- Ensure the added steps are logical, clear, and provide necessary
explanation of the solution process
- If the answer already has explanation, reorganize the solution
into a clear and well-structured format for better readability and
understanding
- For final answers that need exact matching (multiple-choice,
calculations, fill-in-the-blank, true/false), use $\\boxed{}$
notation

Requirements:
- The refined question should include all necessary information
- The refined answer should be accurate and well-explained
- Both question and answer should stand alone (no references to
documents or original materials)

Output format:
First provide your reasoning for the refinements, then output the
final results in this exact format without any notes:

Refined Question: <refined question>
Refined Answer: <refined solution>

I will provide you with the reference document, original question
and its answer. Please analyze them carefully before refinement.

The reference document:
`<DOCUMENT>`

The question:
`<PROBLEM>`

The answer:
`<ANSWER>`
```

Figure 20: Prompt for refining Q-A pairs.

```
You are an expert evaluator tasked with determining whether an
answer contains detailed reasoning processes or explanations of
reasons.

**Task**: Given a question and its corresponding answer, analyze
whether the answer includes:
- Step-by-step reasoning or logical progression
- Detailed explanations of why something is the case
- Cause-and-effect relationships
- Evidence or justifications for conclusions
- Problem-solving methodology or thought processes

**Instructions**:
1. Carefully read both the question and answer
2. Look for explicit reasoning indicators such as:
   - "Because..." / "Since..." / "Therefore..."
   - Sequential steps (First, Second, Then...)
   - Explanatory phrases ("This is due to...", "The reason is...")
   - Logical connectors and transitions
   - Supporting evidence or examples that explain the reasoning
3. Distinguish between mere factual statements and explanatory
reasoning
4. Consider the depth and detail of any reasoning provided

**Output Format**:
Analysis: [Provide your detailed analysis of whether and how the
answer demonstrates reasoning or explanation]
Decision: [YES/NO]

**Examples**:

**Example 1:**
Question: Why does ice float on water?
Answer: Ice floats because it is less dense than water. When water
freezes, its molecules form a crystalline structure that takes up
more space, making ice about 9% less dense than liquid water.

Analysis: The answer provides a clear causal explanation with
scientific reasoning. It explains the mechanism (molecular
structure change) and quantifies the density difference, showing
detailed reasoning about why the phenomenon occurs.
Decision: YES

**Example 2:**
Question: What is the capital of France?
Answer: Paris.

Analysis: This is a simple factual answer without any reasoning
process or explanation. It directly states the fact but provides no
reasoning about why Paris is the capital or any explanatory
context.
Decision: NO

Now analyze the following:
Question:
`<PROBLEM>`

Answer:
`<ANSWER>`
```

Figure 21: Prompt for identifying answers that lack reasoning processes.

```
You are tasked with filtering QA (Question-Answer) data to identify
problematic entries that should be excluded from a dataset. Please
evaluate the provided question and answer pair and determine if it
should be filtered out.

## Filtering Criteria

Filter out (mark as NO) any QA pairs that have the following
issues:

### 1. Contradictory Answers
The answer contains internal contradictions or conflicting
statements.
**Example:**
- Question: What is 2 + 2?
- Answer: First, 2 + 2 = 4. However, using a different method, 2 +
2 = 5. The correct answer is 4.

### 2. External References
The question references external materials that are not provided,
such as:
- Specific equations by number (e.g., "equation (8.75)")
- Figures or diagrams (e.g., "as shown in Fig. 4-16")
- External documents or sources not included in the context
**Examples:**
- Question: Solve equation (3.14) using the given parameters.
- Question: Based on Figure 2.1, calculate the area of the
triangle.

### 3. Missing or Invalid Answers
The answer does not provide a substantive response to the question,
such as:
- Only stating "None of the above" without proper explanation
- Providing no actual answer to the question asked
- Giving completely irrelevant responses
**Example:**
- Question: What is the capital of France?
- Answer: The correct answer is None of the above. This question
cannot be answered properly.

## Output Format

After evaluating the question and answer pair, provide your
analysis and decision in the following format:

Analysis:
<Provide a brief explanation of your evaluation, noting any issues
found or confirming the QA pair is acceptable>

Decision:
<YES/NO>

- YES: Keep this QA pair (it passes the filtering criteria)
- NO: Filter out this QA pair (it has one or more of the issues
listed above)

The question:
`<PROBLEM>`

The answer:
`<ANSWER>`
```

Figure 22: Prompt for filtering defective Q-A pairs.

```
I will now give you two questions: Original question and Candidate
question. Please help me determine if the following two questions
are the same.

Original question:
`<ORIGINAL_PROBLEM>`

Candidate question:
`<CANDIDATE_PROBLEM>`

Disregard the names and minor changes in word order that appear
within.
If their question prompts are very similar and, without considering
the solution process, they produce the same answer, we consider
them to be the same question.

Output Format:
Analysis: [Provide a detailed analysis evaluating the similarity
between these questions]
Decision: [YES/NO]
```

Figure 23: LLM prompt for decontamination.

```
## Task Description
You are tasked with extracting the final reference answer from a
detailed solution that contains both reasoning steps and the final
answer. The reference answer should be concise and represent the
definitive conclusion that can be used as a standard solution.

## Input Format
You will receive:
1. A question that was asked
2. A detailed answer that includes reasoning steps and the final
answer

## Output Requirements
- Extract ONLY the final reference answer without the reasoning
steps
- Ensure the reference answer is complete and can stand alone
- Format the reference answer clearly and concisely
- Do not add any additional explanations or reasoning not present
in the original answer
- If multiple possible answers are given, identify the one marked
as final or preferred

## Example
### Question:
What is the area of a circle with radius 5 cm?

### Detailed Answer:
To find the area of a circle, I need to use the formula A = πr².
Given information: radius = 5 cm
Substituting values: A = π × 5² = π × 25 = 78.54 cm²
Therefore, the area of the circle with radius 5 cm is 78.54 cm².

### Reference Answer:
78.54 cm²

## Instructions
1. Read the question carefully to understand what is being asked
2. Analyze the detailed answer to identify where the final
conclusion is stated
3. Extract only the reference answer without any additional
reasoning
4. Format the reference answer clearly so it can be used for
checking solutions

## Question:
`<PROBLEM>`

## Detailed Answer:
`<ANSWER>`

Now process and return the result.
```

Figure 24: Prompt for annotating reference answer.

```
You are an experienced education evaluator tasked with assessing
student responses to academic questions. Your goal is to analyze
each student answer in relation to the reference answer and provide
both detailed feedback and a numerical score.

Evaluation Process:
1. Carefully read the question to understand the specific
requirements and expected knowledge being tested.
2. Compare the student's response to the reference answer, focusing
on:
    - Conceptual understanding
    - Accuracy of information
    - Completeness of the answer
    - Use of appropriate terminology
    - Logical reasoning and structure
    - Mathematical correctness (where applicable)

3. Provide a thorough analysis that:
    - Identifies specific strengths in the student's response
    - Points out any errors, misconceptions, or omissions
    - Evaluates how well the answer addresses all parts of the
    question
    - Considers whether the student demonstrated the required
    knowledge and skills

4. Assign a score on a scale of 0-10 where:
    - 0: No relevant content or completely incorrect
    - 1-3: Major conceptual errors or significant omissions
    - 4-5: Partial understanding with notable gaps
    - 6-7: Good understanding with minor errors or omissions
    - 8-9: Strong grasp of concepts with minimal errors
    - 10: Complete and perfect answer matching the reference answer

Special Considerations:
- For intervals/ranges: The student's answer must cover the EXACT
SAME range as the reference answer
- For multiple solutions: If the reference answer contains multiple
solutions (connected by "or"/"and"), all must be present in the
student's answer
- For mathematical proofs or procedural questions: Evaluate both
the final answer and the method used
- For conceptual questions: Focus on the depth of understanding and
clarity of explanation

Your response must always follow this format:
Reasoning: <Provide detailed analysis of the student's answer in
relation to the reference answer>
Score: <numerical score between 0 and 10>

The question:
`<PROBLEM>`

The reference answer:
`<REFERENCE_ANSWER>`

The student's answer:
`<STUDENT_ANSWER>`
```

Figure 25: Prompt for evaluating model responses against reference answers

