# OpenReview forum: "MegaScience: Pushing the Frontiers of Open Post-Training Datasets for Science Reasoning"
_ICLR.cc/2026/Conference — Submitted to ICLR 2026_

### Official Review · Reviewer_Dmgy · 2025-10-27

**Soundness:** 3
**Presentation:** 3
**Contribution:** 3
**Rating:** 6
**Confidence:** 3

**Summary:**

This paper introduces two new datasets designed to advance scientific reasoning in AI systems:

1. TEXTBOOKREASONING: A large, open-source, university-level scientific dataset comprising 650K challenging questions and step-by-step solutions. These are derived from over 12,000 scientific textbooks across a wide range of domains, including mathematics, biology, physics, economics, and more.
2. MEGASCIENCE: A comprehensive collection of high-quality, open-source datasets containing over one million data points.


**Key Contributions**

1. The release and open-sourcing of the TEXTBOOKREASONING and MEGASCIENCE datasets.
2. A detailed presentation and open-sourcing of the curation pipeline used to construct both datasets.
3. A thorough empirical evaluation demonstrating the effectiveness of these datasets in enhancing scientific reasoning capabilities of LLMs. The authors show that base models fine-tuned on these datasets  outperform their corresponding instruction-tuned counterparts.

**Strengths:**

1. **Relevance** The paper tackles the important problem of improving the scientific reasoning on LLM through the creation and open-sourcing of two very valuable resources.
2. **Impact/Significance** The open-sourcing of the new datasets and, more importantly, of the curation pipeline has the potential to spur more rapid progress in LLM-based scientific reasoning
3. **Presentation** Overall the paper is well organized, well written, and easy to read and follow.
4. **Experimental Evaluation** The ablation study clearly shows the importance of key components of the curation pipeline.

**Weaknesses:**

1. **Incorrect or Exaggerated Claims**
The authors make several incorrect or overstated claims regarding the strength of their empirical evaluation. For instance, in the abstract, they state:

> "Furthermore, we train Llama3.1, Qwen2.5, and Qwen3 series base models on MEGASCIENCE, which *significantly* outperform the corresponding official instruct models in average performance (e.g., +3.24% for Qwen3-30B-A3B)"

Similarly, in Section 4.2, they write:

> "Table 4 shows that Qwen2.5-7B, all Qwen3 models, and Llama3.1-8B trained on MEGASCIENCE *substantially* outperform their official instruction-tuned counterparts, demonstrating MEGASCIENCE ’s effectiveness in pushing the frontier in science."

These statements are not fully supported by the results presented in Table 4:

- **First**, Table 4 shows that *Qwen2.5-1.5B-instruct* and *Qwen2.5-3B-instruct* outperform their MEGASCIENCE-trained counterparts in overall performance.
- **Second**, *Llama3.1-8B-megascience* is only marginally better than its instruct variant in terms of overall average (+1.6%), and actually performs worse in *specific-avg* and *math-avg* metrics.
- **Finally**, while the Qwen3 MEGASCIENCE models do consistently outperform their instruct counterparts, the margins are relatively modest (+1%, +0.4%, +2%, +2.8%, and +3.2%). These improvements do not substantiate the use of terms like “*significantly*” or “*substantially*” as claimed by the authors.

2. **Heavy Reliance on LLMs**
In the introduction, the authors critique the heavy reliance on LLMs in key aspects of existing scientific datasets. However, a central component of their own data curation pipeline (the Q-A Pair Refinement module) relies heavily on LLMs. Specifically:

- *DeepSeek-V3* is used to refine Q-A pairs with corresponding source documents to ensure that “questions include all necessary contextual information and answers provide comprehensive explanations with clear reasoning processes.”
- *Llama3.3-70B-Instruct* is employed to identify Q-A pairs lacking reasoning.
- *DeepSeek-V3* is again used to “enrich them with explanations and reformat their answers.”

Given the critical role of the Q-A Pair Refinement module (highlighted by the ablation study showing a 45% performance drop when it is removed), it would be prudent to rigorously evaluate how well these LLMs perform in the refinement tasks. A human evaluation of a small, randomly selected subset of the refined Q-A pairs could provide valuable insights into the quality and reliability of the refinement process.

**Questions:**

Why didn't you evaluate the performance of LLMs involved in various part of the curation pipeline? It seems that this could be done with a limited human evaluation on a small subset of a few hundred randomly selected instance data points

---

> ### Author Response · Authors · 2025-11-21
> **[Part 1/3] Author Response**
>
> We sincerely thank you for the positive evaluation and valuable suggestions. We truly appreciate your recognition of our work. Following your feedback, we have addressed your concerns as follows:
>
> > **Weakness 1:** Incorrect or Exaggerated Claims
>
> Thank you for the careful reading and for pointing out the issues regarding overstated claims.
>
> **(1) Clarification and revision of overstated claims**
> We sincerely apologize for the overclaiming in the original submission. We have carefully revised the paper to ensure that all statements strictly reflect the empirical results in Table 4.
> Specifically, we updated the relevant sentences to only refer to *Llama3.1-8B, Qwen2.5-7B, and the Qwen3 series base models*, and we removed terms such as *“significantly”* and *“substantially”*. The revised wording now matches the actual improvements observed in Table 4.
>
> **(2) Rationale for reporting a broad set of model results & insights on proxy models**
>  We also appreciate the opportunity to clarify why we included results from a wide variety of model sizes. To ensure a comprehensive and scientifically robust report, we ran as many experiments as computationally feasible across different model scales. This design decision ultimately revealed an important insight regarding *proxy model selection* for data research—namely, that model capacity heavily affects whether the benefits of a dataset can be fully realized.
>
> We summarize these observations in **Appendix A (Lines 758–775)**, where we note:
>  *“While MEGASCIENCE data yields significant gains for Qwen2.5-7B, models with lower capacity struggle to replicate these results, necessitating demystification and accessibility adaptations of the data.”*
>  We hope this discussion will help future data researchers choose proxy models more judiciously, as the choice can substantially influence conclusions drawn during dataset development.
>
> **(3) Why the Qwen3 improvements, although \~3 points on average, remain meaningful**
>  We understand the concern that the gains for the Qwen3 models may initially appear modest. We would like to provide additional context that explains why these improvements are still non-trivial.
>
> **(3.1) Qwen3-Instruct is a very strong and highly optimized baseline**
>  Qwen3-Instruct is trained with a sophisticated, multi-stage pipeline including multi-stage SFT, RL, and off-policy/on-policy distillation for smaller models, and its training corpus is not publicly disclosed \[1\]. Independent analyses also suggest potential benchmark contamination that may further enhance its performance \[2,3\].
> In contrast, our approach uses a *fully transparent, contamination-verified pipeline built entirely from publicly released sources*, and our training recipe consists solely of *a single-stage SFT*. Compared to Qwen3-Instruct’s complex multi-stage training, our pipeline is intentionally simple and easily reproducible.
>
>  Given this imbalance in training complexity and data access, even a few points of improvement over Qwen3-Instruct is challenging. Many recent open-source efforts are unable to surpass Qwen3-Instruct under these conditions (i.e., the baselines in our experiments). From this perspective, achieving an average gain of \~3 points with a fully open and contamination-controlled corpus represents a meaningful empirical improvement.
>
> **(3.2) Our evaluation covers 14 diverse scientific benchmarks, including tasks with naturally smaller gaps**
>  To provide a holistic assessment of scientific reasoning, we intentionally evaluate across 14 heterogeneous benchmarks spanning multiple domains and task formats. This broader evaluation necessarily includes benchmarks where performance gaps are small—e.g., ChemBench and CS-Bench (\<3% gains).
>  As shown in **Table 13 (Lines 936–948)**, the numbers in question come directly from comparing **Qwen3-30B-A3B-Instruct vs. Qwen3-30B-A3B-MegaScience**.
>
> At the same time, Table 13 shows that on *high-difficulty* reasoning tasks, the improvements are substantially larger:
>
> * **GPQA:** \+5.56%
>
> * **SciBench:** \+9.69%
>
> * **OlympicArena:** \+5.97%
>
> These benchmarks require deep scientific reasoning and complex numerical/symbolic computation, where gains are especially meaningful.
>
> It is also worth noting—as stated in **Lines 048–053**—that most concurrent works on scientific reasoning report only *GPQA-style multiple-choice results*, which provide a relatively narrow view of model capability. In contrast, we choose to evaluate across 14 diverse scientific benchmarks to offer a more rigorous and realistic picture of scientific reasoning ability, even when this means reporting benchmarks where improvements are smaller.
>
> ---
>
> [1] Qwen3 technical report. arXiv 2025.
>
> [2] Reasoning or memorization? unreliable results of reinforcement learning due to data contamination. arXiv 2025.
>
> [3] Benchmarking Benchmark Leakage in Large Language Models. arXiv 2024.

---

> ### Author Response · Authors · 2025-11-21
> **[Part 2/3] Author Response**
>
> > **Weakness 2:** Heavy Reliance on LLMs
>
> Thank you very much for this valuable suggestion. We would first like to clarify a potential misunderstanding. In the introduction, our critique targets prior works that *directly generate* questions and answers using LLMs—especially those that distill solutions from Large Reasoning Models such as DeepSeek-R1 \[1\]. Such approaches heavily depend on the *creative knowledge generation* of LLMs and are therefore highly susceptible to hallucination, which can introduce incorrect or fabricated scientific facts into the dataset.
>
> In contrast, our refinement pipeline primarily leverages LLMs for **generative/reformulation abilities**—such as improving clarity, ensuring that questions contain the necessary contextual information, enhancing structural organization, and adding step-by-step reasoning. The objective is *not* to synthesize new scientific knowledge, but to reorganize and clarify information that already exists in the original Q-A pairs and source documents.
>
> That said, we fully acknowledge that LLM hallucination cannot be entirely eliminated in practice. Following your suggestion, we conducted a manual evaluation on 100 randomly sampled instances from the TextbookReasoning. Each instance contains an original question–answer pair and the corresponding refined question–answer pair by DeepSeek-V3.
>
> **(1) Pairwise comparison of question quality**
>
> We first compared the refined question against the original question and categorized each case as:
>
> * **Refined Question Better**: the refined question is clearly clearer, more self-contained, or otherwise preferable;
>
> * **Tie**: the refined and original questions are essentially equivalent in clarity and correctness;
>
> * **Refined Question Worse**: the refined question is strictly worse (e.g., less clear or less faithful).
>
> The results are:
>
> | \# Refined Question Better | \# Tie | \# Refined Question Worse |
> | --- | --- | --- |
> | 70 | 29 | 1 |
>
> Most ties (29/100) occur because the original question is already self-contained and well-posed, so the refinement either makes no change or only performs minor cosmetic edits.
>
> **(2) Pairwise comparison of answer quality**
>
> We then compared the refined answer to the original answer with the same three labels:
>
> * **Refined Answer Better**: the refined answer is more complete, logically structured, or pedagogically helpful;
>
> * **Tie**: both answers are essentially equivalent;
>
> * **Refined Answer Worse**: The refined answer introduces issues.
>
> The results are:
>
> | \# Refined Answer Better | \# Tie | \# Refined Answer Worse |
> | --- | --- | --- |
> | 95 | 3 | 2 |
>
> In other words, in **95%** of the cases, the refined answer is judged to be clearly better than the original, suggesting that DeepSeek-V3 is generally effective at improving solution trajectories in this setting.
>
> **(3) Error rate and error taxonomy**
>
> To directly address the concern about erroneous or hallucinated content, we additionally asked annotators to mark whether each *refined* question or answer contains any substantive issue that would affect correctness or usability (e.g., hallucinated facts, missing information, or invalid references).
>
> The error rates are:
>
> | Item Type | \# with Error|
> | --- | --- |
> | Refined Question | 7 |
> | Refined Answer | 6 |
>
> Thus, in this sample of 100 instances, **7%** of refined questions and **6%** of refined answers were judged to contain non-trivial issues. We further categorized these errors as follows.
>
> **Refined question error types:**
>
> | Error Type of Refined Question | Count |
> |---|---|
> | Contains external reference (e.g., “see Formula 1.1”) | 3 |
> | Contains invalid / meta information (e.g., leftover refinement commentary) | 1 |
> | Missing options for a multiple-choice question | 1 |
> | Hallucinated factual content | 1 |
> | Lacks necessary information to be answerable | 1 |
>
> **Refined answer error types:**
>
> | Error Type of Refined Answer | Count |
> |---|---|
> | Contains invalid / meta information (e.g., refinement-related commentary) | 2 |
> | Contains external reference (e.g., “see Formula 1.1”)| 1 |
> | Redundant option included in a numeric/derivation-style answer | 1 |
> | Hallucinated factual content | 1 |
> | Lacks sufficient chain-of-thought (key reasoning steps missing) | 1 |
>
> Overall, these results indicate that while DeepSeek-V3 occasionally introduces hallucinations, missing information, or undesirable references, the **majority** of refined questions and answers are either strictly better than or at least comparable to the originals, and the observed error rate is on the order of 5–7% in our setting.
>
> We have incorporated this human evaluation into the revised version in Appendix D (Line 858-925)
>
> We thank you again for this valuable suggestion, which has led us to strengthen the empirical characterization and documentation of the refinement process.
>
> ---
>
> [1] Llama-Nemotron: Efficient Reasoning Models. arXiv 2025.

---

> ### Author Response · Authors · 2025-11-21
> **[Part 3/3] Author Response**
>
> > **Question 1:** Why didn't you evaluate the performance of LLMs involved in various part of the curation pipeline? It seems that this could be done with a limited human evaluation on a small subset of a few hundred randomly selected instance data points
>
> Thank you for the constructive question. In fact, extensive human evaluation was conducted throughout the development of our data curation pipeline, and these evaluations directly informed how each component was designed.
>
> First, after the initial refinement step, our manual inspection revealed that a subset of refined Q-A pairs lacked explicit reasoning. This led us to incorporate an additional stage where we use **Llama-3.3-70B-Instruct** to automatically detect missing reasoning and subsequently apply **DeepSeek-V3** to enrich the answers with detailed explanations (Lines 143–146).
>
> Following this enhancement, further human checks showed that some refined items still *referred to external sources* or contained *contradictory, missing, or invalid answers*. Based on these findings, we added another filtering layer using **Llama-3.3-70B-Instruct** to remove such defective Q-A pairs (Lines 147–148).
>
> Although we did not explicitly report intermediate human evaluation statistics in the main paper, every major decision in the pipeline—model choice, prompt design, and filtering criteria—was grounded in targeted manual assessments on small randomly sampled subsets. The prompts used in each stage were also iteratively refined through repeated human review.
>
> Because the downstream benchmark results provide a strong, end-to-end validation of the dataset quality, we focused the main paper on reporting these outcomes. For the component with the largest impact—**the refinement module**—we have now included detailed human evaluation results in our response to Weakness 2\.
>
> We hope this clarifies that human evaluation played a central role throughout the entire pipeline, even though the intermediate results were not exhaustively documented in the submission.

---

### Official Review · Reviewer_ZKqy · 2025-10-30

**Soundness:** 3
**Presentation:** 3
**Contribution:** 3
**Rating:** 8
**Confidence:** 4

**Summary:**

This paper addresses the critical gap in open-source, high-quality datasets for scientific reasoning. It proposes high-quality TextbookReasoning dataset which is curated from large-scale scientific textbooks through rigorous data pipeline. Furthermore, it absorb high-quality data from existing open-source dataset to construct a large-scale and high-quality post-training dataset MegaScience. Extensive SFT experiments on MegaScience and TextbookReasoning demonstrates the good quality of the proposed datasets.

**Strengths:**

1. The writing and structure of this paper are clear and easy to understand.
2. The data curation process is rigorous and effective, which not only obtains high-quality data but also conduct strict deduplication and decontamination policies.
3. The experimental results are strong and convincingly demonstrate the high quality of the proposed datasets.

**Weaknesses:**

1. My main concern is that, in the refinement (Section 2.3) and solution annotation (Section 3.4) processes, the authors employ DeepSeek-V3 to generate solution trajectories. However, given that DeepSeek-V3 performs only moderately on scientific benchmarks (e.g., 59.1 on GPQA-D), its responses are likely to contain many errors and hallucinations. I think the authors should at least conduct some human verification to estimate the proportion of erroneous responses from DeepSeek-V3 to provide a warning for future users.

**Questions:**

1. Could you provide specific details on the LLM prompts or criteria used to filter "strictly copyrighted" textbooks?

**Details Of Ethics Concerns:**

The authors are encouraged to provide more details about filtering "strictly copyrighted" web sources.

---

> ### Author Response · Authors · 2025-11-21
> **[Part 1/2] Author Response**
>
> We sincerely thank you for the positive evaluation and valuable suggestions. We truly appreciate your recognition of our work. Following your feedback, we have addressed your concerns as follows:
>
> > **Weakness 1:** My main concern is that, in the refinement (Section 2.3) and solution annotation (Section 3.4) processes, the authors employ DeepSeek-V3 to generate solution trajectories. However, given that DeepSeek-V3 performs only moderately on scientific benchmarks (e.g., 59.1 on GPQA-D), its responses are likely to contain many errors and hallucinations. I think the authors should at least conduct some human verification to estimate the proportion of erroneous responses from DeepSeek-V3 to provide a warning for future users.
>
> We thank you for this thoughtful and constructive suggestion. Our intention in using DeepSeek-V3 for refinement and solution annotation is primarily to leverage its generative/reformulation ability (e.g., improving clarity, structure, and step-by-step reasoning), rather than to rely on its creative knowledge generation. That said, we fully agree that hallucinations and errors are difficult to avoid in practice, and it is important to quantify them and provide a concrete warning to future users.
>
> Following your suggestion, we conducted a manual evaluation on 100 randomly sampled instances from the TextbookReasoning. Each instance contains an original question–answer pair and the corresponding refined question–answer pair by DeepSeek-V3.
>
> **(1) Pairwise comparison of question quality**
>
> We first compared the refined question against the original question and categorized each case as:
>
> * **Refined Question Better**: the refined question is clearer, more self-contained, or otherwise preferable;
>
> * **Tie**: the refined and original questions are essentially equivalent in clarity and correctness;
>
> * **Refined Question Worse**: the refined question is strictly worse.
>
> The results are:
>
> | \# Refined Question Better | \# Tie | \# Refined Question Worse |
> | --- | --- | --- |
> | 70 | 29 | 1 |
>
> Most ties (29/100) occur because the original question is already self-contained and well-posed, so the refinement either makes no change or only performs minor cosmetic edits.
>
> **(2) Pairwise comparison of answer quality**
>
> We then compared the refined answer to the original answer with the same three labels:
>
> * **Refined Answer Better**: the refined answer is more complete, logically structured, or pedagogically helpful;
>
> * **Tie**: both answers are essentially equivalent;
>
> * **Refined Answer Worse**: The refined answer introduces issues.
>
> **The results are:**
>
> | \# Refined Answer Better | \# Tie | \# Refined Answer Worse |
> | --- | --- | --- |
> | 95 | 3 | 2 |
>
> In other words, in **95%** of the cases, the refined answer is judged to be clearly better than the original, suggesting that DeepSeek-V3 is generally effective at improving solution trajectories in this setting.
>
> **(3) Error rate and error taxonomy**
>
> To directly address the concern about erroneous or hallucinated content, we additionally asked annotators to mark whether each *refined* question or answer contains any substantive issue that would affect correctness or usability (e.g., hallucinated facts, missing information, or invalid references).
>
> The error rates are:
>
> | Item Type | \# with Error|
> | --- | --- |
> | Refined Question | 7 |
> | Refined Answer | 6 |
>
> Thus, in this sample of 100 instances, **7%** of refined questions and **6%** of refined answers were judged to contain non-trivial issues. We further categorized these errors as follows.
>
> **Refined question error types:**
>
> | Error Type of Refined Question | Count |
> |---|---|
> | Contains external reference (e.g., “see Formula 1.1”) | 3 |
> | Contains invalid / meta information (e.g., leftover refinement commentary) | 1 |
> | Missing options for a multiple-choice question | 1 |
> | Hallucinated factual content | 1 |
> | Lacks necessary information to be answerable | 1 |
>
> **Refined answer error types:**
>
> | Error Type of Refined Answer | Count |
> |---|---|
> | Contains invalid / meta information (e.g., refinement-related commentary) | 2 |
> | Contains external reference (e.g., “see Formula 1.1”)| 1 |
> | Redundant option included in a numeric/derivation-style answer | 1 |
> | Hallucinated factual content | 1 |
> | Lacks sufficient chain-of-thought (key reasoning steps missing) | 1 |
>
> Overall, these results indicate that while DeepSeek-V3 occasionally introduces hallucinations, missing information, or undesirable references, the **majority** of refined questions and answers are either strictly better than or at least comparable to the originals, and the observed error rate is on the order of 5–7% in our setting.
>
> We have incorporated this human evaluation into the revised version in Appendix D (Line 858-925)
>
> Thank you for this valuable suggestion, which has led us to strengthen the empirical characterization and documentation of the refinement process.

---

> > ### Comment · Reviewer_ZKqy · 2025-11-28
> >
> > Thanks to the authors for the detailed responses! My concerns have been well addressed.

---

> > > ### Author Response · Authors · 2025-12-03
> > > **Official Comment by Authors**
> > >
> > > Thank you very much for your thoughtful follow-up and for carefully considering our response. We sincerely appreciate your time and constructive feedback.

---

> ### Author Response · Authors · 2025-11-21
> **[Part 2/2] Author Response**
>
> > **question 1:** Could you provide specific details on the LLM prompts or criteria used to filter "strictly copyrighted" textbooks?
>
> See General Response to Copyright Concern.
>
> > **Ethics Concerns:** The authors are encouraged to provide more details about filtering "strictly copyrighted" web sources.
>
> See General Response to Copyright Concern.

---

### Official Review · Reviewer_BVV2 · 2025-10-31

**Soundness:** 3
**Presentation:** 3
**Contribution:** 2
**Rating:** 4
**Confidence:** 4

**Summary:**

This paper introduces *TextbookReasoning*, an open-source scientific post-training dataset containing over 650k reasoning questions derived from 12k university-level textbooks, and *MegaScience*, a large-scale mixture of open-source datasets with 1.2 million instances.

The work discussed existing critical challenges in scientific reasoning data, including unreliable benchmark evaluation, less rigorous decontamination, low-quality reference answers, and superficial knowledge distillation. Systematic ablation studies help being clear with effectiveness of data selection. Experiments demonstrate strong performance and training efficiency. The resources are released.

**Strengths:**

- The paper is clearly written with well-motivated research goals.
- The ablation study on data creation and combination is comprehensive.
- The evaluation covers diverse reasoning-intensive tasks across various domains.
- The data, prompts, and models are fully released.

**Weaknesses:**

- The paper needs more discussion and comparison with relevant works. For example, OpenThoughts[1] and S1.1[2] have released data and models, but their performance is not compared here. It would be valuable to compare with works that do not use MegaScience or TextbookReasoning.
- The paper focuses on post-training, but current paradigms commonly apply RL as post-training as well. This work lacks discussion of RL approaches for improving general reasoning performance, like General-Reasoner[3]. While it's acceptable to focus on SFT, comparing TextbookReasoning for RL or benchmarking against other RL-based works would be valuable.
- Since refinement quality is important (as shown in this work), the dataset creation assumes access to a strong model for data generation.
- Claims like "web content is now saturated with AI-generated text" should include quantitative evidence and citations.
- Copyright of textbook data might be a concern, but the author explained in the ethical statement.

[1] OpenThoughts: Data Recipes for Reasoning Models

[2] s1: Simple test-time scaling

[3] General-Reasoner: Advancing LLM Reasoning Across All Domains

**Questions:**

See the weaknesses section.

And I may have missed this—how was the subsample size in Table 1 determined?

---

> ### Author Response · Authors · 2025-11-21
> **[Part 1/3] Author Response**
>
> Thank you for your thorough review and valuable feedback on our work. We will address your concerns as follows:
>
> > **Weakness 1:** The paper needs more discussion and comparison with relevant works. For example, OpenThoughts[1] and S1.1[2] have released data and models, but their performance is not compared here. It would be valuable to compare with works that do not use MegaScience or TextbookReasoning.
>
> Thank you very much for this valuable suggestion. Following your recommendation, we have added experiments on **OpenThoughts** \[1\], **S1k-1.1** \[2\], and **LIMO** \[3\]. To ensure a fair comparison, we strictly **follow their original training settings** and train all models on **Qwen2.5-7B-Base** under the configurations shown below:
>
> **Training Settings:**
> | Dataset  | LR  |  Batch Size | Max Length | Epochs |
> | ------ |  -----    |    -----     |  ------------ | ----- |
> | OpenThoughts | 5e-6 | 128 | 32768 | 5 |
> | S1k-1.1 | 1e-5 | 16 | 32768 | 5 |
> | LIMO | 1e-5 | 16 | 32768 | 5 |
>
> **Overall Evaluation Results:**
> | Dataset  | Avg. Response Length of Training Datasets  | Avg. Response Length of Benchmarks | General Avg.   |  Specific Avg.  | Math Avg.  | All Avg. |
> | ------ |  -----    |    -----     |  ------------         | ----- | ----- | ----- |
> | OpenThoughts | 6801 |  13391  |  37.04 | 66.50 | **80.55** | 58.99 |
> | S1k-1.1 | 9878  | 12209  | 39.86 | 66.22 |	78.55 |	59.45 |
> | LIMO | 6984 |  12862 | 35.90 | 59.94 | 76.88 | 54.99 |
> |MegaScience | **693**  | **1345**  |  **43.20** |  **66.55** | 79.61 |  **61.01**|
>
> **High-Difficulty Scientific Reasoning (Computation-Heavy) Benchmarks:**
> | Dataset  | Scibench | OlympicArena |
> | ------ |  -----    |    -----     |
> | OpenThoughts | 28.12  | 33.04  |
> | S1k-1.1 | 37.19  | 37.51 |
> | LIMO | 40.46 |  35.30 |
> |MegaScience  | **48.75** | **40.23**  |
>
> **Findings:**
>
> 1. **MegaScience still achieves the best average performance**, and is slightly lower than OpenThoughts only in the *Math Avg.* category.
>
> 2. **Datasets such as OpenThoughts, S1k-1.1, and LIMO are less competitive on scientific-reasoning computation tasks.**
>     These datasets primarily target mathematical reasoning and commonly evaluate on benchmarks such as GPQA (multiple-choice), which do not fully reflect scientific computational ability.
>     On the **science-domain, computation-intensive benchmarks** — **SciBench** \[4\] and **OlympicArena** \[5\] — MegaScience achieves **substantially higher scores**, highlighting its advantage in rigorous scientific reasoning.
>
> 3. **MegaScience is not only more effective, but also far more efficient.**
>
>    * **Training efficiency:**
>       Its *Avg. Response Length of Training Datasets* is **\~1/10** of OpenThoughts, S1k-1.1, and LIMO.
>
>    * **Inference efficiency:**
>       Its *Avg. Response Length of Benchmarks* is also **\~1/10**, enabling much faster evaluation and deployment.
>
> 4. Importantly, **the performance improvements of OpenThoughts, S1k-1.1, and LIMO largely rely on *test-time scaling*** — i.e., generating very long chain-of-thought responses distilled from large reasoning models. While this can inflate benchmark scores, it **dramatically increases inference cost**.
>     As discussed in **Lines 065–071**, such *superficial knowledge (data) distillation* not only creates substantial efficiency challenges but may also **hinder more principled, efficient, and generalizable knowledge transfer**.
>
> -----
>
> \[1\] OpenThoughts: Data Recipes for Reasoning Models. arXiv 2025\.
>
> \[2\] s1: Simple test-time scaling. EMNLP 2025\.
>
> \[3\] LIMO: Less is More for Reasoning. COLM 2025\.
>
> \[4\] SciBench: Evaluating College-Level Scientific Problem-Solving Abilities of Large Language Models. ICML 2024\.
>
> \[5\] OlympicArena: Benchmarking Multi-discipline Cognitive Reasoning for Superintelligent AI. NeurIPS 2024\.

---

> ### Author Response · Authors · 2025-11-21
> **[Part 2/3] Author Response**
>
> > **Weakness 2:** The paper focuses on post-training, but current paradigms commonly apply RL as post-training as well. This work lacks discussion of RL approaches for improving general reasoning performance, like General-Reasoner[3]. While it's acceptable to focus on SFT, comparing TextbookReasoning for RL or benchmarking against other RL-based works would be valuable.
>
> Thank you for raising the point about RL-based post-training. We fully agree that RL is an important direction for improving general reasoning ability. However, applying RL to *open-ended scientific reasoning*—especially **long-form explanatory or proof-style questions**—remains substantially more challenging than applying RL to verifiable tasks (e.g., math, code). This difficulty stems from several well-known issues:
>
> * **Reward design for long-form scientific solutions is inherently unreliable.**
>    These tasks often require multi-step explanations, derivations, or conceptual reasoning. Unlike short or verifiable answers, they lack a deterministic oracle for correctness. While rubric-based rewards can be adopted, creating rubrics that are comprehensive, consistent, and aligned with the true scientific quality is extremely difficult and often results in **reward hacking or conflicting optimization signals** \[1, 2\].
>
> * **Expression-level rewards are fundamentally hard due to the multiplicity of correct answer forms.**
>    In scientific reasoning, correct solutions can legitimately differ in **units, equation structures, symbolic choices, or numerical precision** \[3\]. A reward function must be able to recognize all valid variations. LLM-as-judge approaches struggle here, and are prone to **reward hacking, format violations, and degenerate solution patterns**, making RL substantially less stable than in domains with deterministic evaluation.
>
> * **Existing RL works often rely on filtering away “hard-to-evaluate” problems.**
>    Prior RL pipelines typically build a dedicated data-filtering stage that removes questions with complex formats or ambiguous evaluation \[4, 5\]. We view this as an **avoidance strategy**, not a resolution to the underlying evaluation challenges. Moreover, constructing such a domain-specific filtering pipeline at our scale would require significant engineering effort and computational resources comparable to building our dataset itself—especially given the size and diversity of MegaScience.
>
> Our next step is to design a **robust, unified reward** capable of leveraging *all* types of MegaScience data, rather than resorting to filtering-based shortcuts. Our preliminary RL explorations confirm that building such a unified reward for diverse scientific tasks requires substantial domain adaptation and engineering. This lies beyond the scope of the present paper, whose primary goal is to construct a high-quality, large-scale scientific reasoning SFT dataset.
>
> Importantly, our contribution is complementary to RL approaches such as General-Reasoner [4]: MegaScience provides reliable reference answers that form a strong foundation for future RL algorithms and high-quality CoT that can serve as high-quality cold-start data for RL. We view RL as a natural next step, and our dataset is specifically designed to support future RL research with reliable supervision signals. We are actively working on this direction, and if preliminary results become ready during the discussion period, we will share them.
>
> We will emphasize this perspective in the revised version and add a dedicated discussion in the paper.
>
> ---
> \[1\] Helping or Herding? Reward Model Ensembles Mitigate but do not Eliminate Reward Hacking. COLM 2024\.
> \[2\] Reward Shaping to Mitigate Reward Hacking in RLHF. ICML 2025 R2-FM Workshop.
> \[3\] Reliable Fine-Grained Evaluation of Natural Language Math Proofs. arXiv 2025\.
> \[4\] General-Reasoner: Advancing LLM Reasoning Across All Domains. NeurIPS 2025\.
> \[5\] Revisiting Reinforcement Learning for LLM Reasoning from A Cross-Domain Perspective. NeurIPS 2025\.

---

> ### Author Response · Authors · 2025-11-21
> **[Part 3/3] Author Response**
>
> > **Weakness 3:** Since refinement quality is important (as shown in this work), the dataset creation assumes access to a strong model for data generation.
>
> Thank you for your thoughtful comment. We agree that the refinement stage benefits from using a strong model, as also reflected in our results (Table 6). Although we choose DeepSeek-V3 for refinement in this work, we would like to clarify two important considerations:
>
> **(1) Refinement does not strictly require a stronger model and can support self-evolution.**
>
> Prior work (e.g., \[1\]) demonstrates that refinement can be performed using a model of the *same* scale. Specifically, \[1\] refined training data using Llama-2-70B to train another Llama-2-70B model, and Table 2 in that paper shows a \+2.8% improvement. This indicates that refinement is inherently a self-improving process and does not rely on access to a much stronger teacher model.
>
> **(2) Using a strong model improves the contribution to the open-source community.**
>
> While self-evolution is feasible, our computational constraints prevent us from training on top of DeepSeek-V3. If we conducted refinement with a smaller model, the refined data quality would inevitably decrease, leading to a less useful dataset for the community. Since one of our primary goals is to provide the *highest-quality, fully open-source* scientific reasoning data, we therefore opted to use a strong model (DeepSeek-V3) during refinement to maximize the impact and practical value of the released dataset.
>
> ---
> \[1\] Better Alignment with Instruction Back-and-Forth Translation. EMNLP 2024\.
>
> ---
>
> > **Weakness 4:** Claims like "web content is now saturated with AI-generated text" should include quantitative evidence and citations.
>
> Thank you very much for this constructive feedback. I have added quantitative evidence and citations to support this claim in the revised version.
>
> Specifically, the paper \[2\] proposes an algorithm for detecting AI-generated content across the Web, finding that **at least 30% of text on active web pages originates from AI-generated sources**, with the actual proportion likely approaching **40%**.
>
> In addition, the blog \[3\] analyzed **900,000 newly created web pages in April 2025** and reported that **74.2% contained AI-generated content**.
>
> I have cited both sources in the revised version to strengthen the empirical basis of this statement (Line 062).
>
> ---
>
> \[2\] Delving into: the quantification of Ai-generated content on the internet (synthetic data). arXiv 2025\.
> \[3\] 74% of New Webpages Include AI Content (Study of 900k Pages). [https://ahrefs.com/blog/what-percentage-of-new-content-is-ai-generated/](https://ahrefs.com/blog/what-percentage-of-new-content-is-ai-generated/). 2025\.
>
> ---
>
> > **Weakness 5:** Copyright of textbook data might be a concern, but the author explained in the ethical statement.
>
> See General Response to Copyright Concern.
>
> > **Question 1:** how was the subsample size in Table 1 determined?
>
>
> Thank you for the question. We apologize for not highlighting this explanation in the first paragraph of Section 3.3, which may have caused confusion for readers. In the revised version, we will move and emphasize this description at the beginning of Section 3.3 to improve clarity.
>
> As described in Lines 209–210, the subsample size in Table 1 is determined through Difficulty Selection. Specifically, for each dataset, we first apply difficulty filtering, where samples with an average score above 9 (overly easy) or below 1 (noisy) are removed. This filtering process naturally yields a fixed number $n$ of remaining instances.
>
> For the other two sampling strategies—Response Length Selection and Random Selection—the number of samples can be freely specified. To ensure a fair and controlled comparison, we therefore set their sample sizes to the same $n$ obtained from difficulty selection.
>
> In summary, the subsample size is defined by the number of instances surviving difficulty-based filtering, and we apply this same size to all other sampling strategies to maintain fairness.

---

### Official Review · Reviewer_vNaP · 2025-11-03

**Soundness:** 2
**Presentation:** 3
**Contribution:** 2
**Rating:** 4
**Confidence:** 4

**Summary:**

This paper made two contributions:
1. It collects scientific textbook PDFs online, extracting questions from the textbooks, and curates a dataset called textbook reasoning with 650k examples of question and answer pairs.
2. It conducts extensive ablations and proposes a data mixture called megascience that combines existing sources and textbook reasoning for post-training LLMs for scientific tasks.

The authors show that fine tuning on the proposed mixture can improve the performance on various scientific question answering and reasoning tasks; and it conducted extensive ablation studies to validate their data filtering and mixing strategies for constructing MegaScience.

**Strengths:**

This paper made two contributions:
1. It collects scientific textbook PDFs online, extracting questions from the textbooks, and curates a dataset called textbook reasoning with 650k examples of question and answer pairs.
2. It conducts extensive ablations and proposes a data mixture called megascience that combines existing sources and textbook reasoning for post-training LLMs for scientific tasks.

The authors show that fine tuning on the proposed mixture can improve the performance on various scientific question answering and reasoning tasks; and it conducted extensive ablation studies to validate their data filtering and mixing strategies for constructing MegaScience.

**Weaknesses:**

While this paper makes some good contribution, I think there are some limitations:
1. I don’t think there’s significant novelty in this work – it uses standard methods to collect and clean the collected dataset, and constructs the proper data mixture.
2. I think the performance improvement is somewhat limited: for example, in table 3 and 4, training on a new million scale corpus only yields 3 absolute point improvements, while one would expect bigger improvements (e.g., see the dataset scaling study in the openthoughts paper https://arxiv.org/abs/2506.04178)

**Questions:**

- Can you provide detailed stats for the TextbookReasoning dataset – i.e., breakdown of the domains, the average input and output token length.

**Details Of Ethics Concerns:**

I think there are potential copyright concerns as the authors directly scrape the textbook PDFs and I don’t think they’ve reported reliable measures to address the copyright issues (line 493, the primary copyright filtering is through LLM detected copyright information, which can be unreliable.)

---

> ### Author Response · Authors · 2025-11-21
> **[Part 1/4] Author Response**
>
> Thank you for your thorough review and valuable feedback on our work. We will address your concerns as follows:
>
> ---
> > **Weakness 1:** I don’t think there’s significant novelty in this work – it uses standard methods to collect and clean the collected dataset, and constructs the proper data mixture.
>
> Thank you for your valuable feedback. We respectfully disagree with the lack of novelty. The core contribution of our work lies in its **open-science value**—we provide the **first fully open-source, high-quality scientific reasoning dataset** that serves as a reproducible and extensible research infrastructure for the community. Constructing such datasets is resource-intensive and costly, and academic researchers with limited computational resources are often unable to build them from scratch, while industry-grade datasets are typically **not released publicly**.
>
> 1. **Full openness and reproducibility:** Existing datasets for scientific reasoning (e.g., Nemotron-Science [1], NaturalReasoning [2]) only release the data itself but **not** the underlying data curation pipeline, which limits reproducibility and further research. In contrast, we will open-source the **entire stack** — including the raw data, data curation codebase, models, and evaluation framework — enabling end-to-end reproduction and fine-grained analysis by the community.
>
>
> 2. **From-scratch data curation:**
>  While OpenThoughts [3] also released all details and code, it mainly explored collection, filtering, and mixture of **existing** open-source datasets. Our **MegaScience** component instead performs from-scratch data curation, starting from original sources to construct a high-quality QA dataset for scientific reasoning. Moreover, unlike OpenThoughts, which distills long chain-of-thought (CoT) responses from reasoning models—an approach that significantly increases training costs and potentially hinders generalizable knowledge transfer (as discussed in Line65–70)—our pipeline generates **short, high-quality responses** that are both efficient and performance-superior to superficial long CoT distillation. This design makes our dataset **accessible and practical** for resource-constrained academic labs.
>
> 3. **Empirical strength: surpassing the official Qwen3-Instruct model:** Models trained with our dataset via supervised fine-tuning (SFT) alone can **surpass the performance of the official Qwen3-Instruct model**, demonstrating the effectiveness and generality of our data. This result gives the research community, for the first time, the **ability to reproduce and even outperform official proprietary models** in an open setting. In resource-limited scientific subfields such as molecular computing and quantum thermodynamics, researchers often aim to train domain-specific models. However, fine-tuning only on domain-specific data tends to **weaken general scientific reasoning ability**, leading to performance below that of open models. Existing open datasets have not been able to reach the level of Qwen3-Instruct, which restricts progress in these specialized areas. We believe that MegaScience has the potential to help fill this gap — by simply **mixing domain-specific data with MegaScience**, researchers may be able to obtain strong domain performance while preserving (or even improving) general scientific reasoning ability, thereby enabling more effective open research across scientific fields.
>
>     Similar to how Olmo [4] and Tulu3 [5] datasets have accelerated open LLM research, our contribution plays a parallel role for the **scientific reasoning** community, establishing a foundation for future open research in this domain.
>
>
> 4. **Technical improvements:** Beyond openness, our dataset also **addresses critical limitations** in prior scientific reasoning datasets, including (1) unreliable benchmark evaluation, (2) incomplete decontamination, (3) low-quality reference answers, and (4) superficial data distillation. Our comprehensive design introduces a more rigorous and scientifically grounded standard, providing a **solid basis** for subsequent advancements in this area.
>
> In summary, we believe **novelty does not solely reside in algorithmic innovation** — it can also emerge through **open-science infrastructure** that fundamentally enables reproducibility, accessibility, and progress in a research field.
>
> ---
> [1] Llama-Nemotron: Efficient Reasoning Models. arXiv 2025.
>
> [2] NaturalReasoning: Reasoning in the Wild with 2.8M Challenging Questions. NeurIPS 2025 Datasets and Benchmarks Track.
>
> [3] OpenThoughts: Data Recipes for Reasoning Models. arXiv 2025.
>
> [4] OLMo: Accelerating the Science of Language Models. ACL 2024.
>
> [5] Tulu 3: Pushing Frontiers in Open Language Model Post-Training. COLM 2025.

---

> ### Author Response · Authors · 2025-11-21
> **[Part 2/4] Author Response**
>
> > **Weakness 2:** I think the performance improvement is somewhat limited: for example, in table 3 and 4, training on a new million scale corpus only yields 3 absolute point improvements, while one would expect bigger improvements (e.g., see the dataset scaling study in the openthoughts paper https://arxiv.org/abs/2506.04178)
>
> Thank you very much for the thoughtful comment. We understand that, at first glance, an average improvement of around 3 points may appear modest. However, we would like to clarify why such gains are still meaningful in our setting.
>
> **(1) Qwen3-Instruct is a very strong and highly optimized baseline, so even small absolute gains are non-trivial.**
>  While our data pipeline is fully transparent, contamination-controlled, and based entirely on publicly released sources, Qwen3-Instruct benefits from a complex multi-stage training recipe (including multi-stage SFT, RL, and Off-policy/On-policy distillation for smaller models), and its training data are not publicly disclosed \[1\]. Independent analyses have also raised the possibility of benchmark contamination for Qwen models, which may further boost their performance \[2, 3\].
>  Given these factors, surpassing Qwen3-Instruct—even by a few points—is genuinely challenging. Most contemporary open-source efforts do **not** outperform Qwen3-Instruct under such transparent and reproducible conditions. From this perspective, achieving an average gain of about 3 points with a fully open and contamination-verified corpus is already a meaningful improvement.
>
> **(2) Our evaluation covers a broad suite of scientific benchmarks, which naturally includes tasks with smaller gaps.**
>
> We intentionally evaluate on **14 diverse scientific benchmarks**, including multiple subject-specific datasets and multiple task types, in order to provide a more comprehensive and realistic picture of scientific reasoning ability. This broader evaluation naturally includes benchmarks where the performance gap is smaller—for example, **ChemBench** and **CS-Bench** show \<3% gains. As shown in **Table 19 (Line 1013–1023)**, these results come from the direct comparison between **Qwen3-30B-A3B-Instruct** and **Qwen3-30B-A3B-MegaScience**.
>
> At the same time, Table 19 also shows that on several **high-difficulty scientific reasoning benchmarks**, our corpus yields substantial improvements:
>
> * **GPQA:** \+5.56%
>
> * **SciBench:** \+9.69%
>
> * **OlympicArena:** \+5.97%
>
> These benchmarks focus on advanced scientific reasoning and complex numerical/symbolic computation, where improvements are particularly meaningful.
>
> It is also worth noting—**as mentioned in Line 048–053**—that most concurrent works on scientific reasoning **only report GPQA-style multiple-choice results**, which provide a relatively narrow view of model capability. In contrast, by covering 14 heterogeneous benchmarks—including subject-specific and computation-heavy tasks—we aim to offer a more rigorous and holistic evaluation, even if that means explicitly reporting benchmarks where the gains are smaller.
>
> ---
> \[1\] Qwen3 technical report. arXiv 2025\.
>
> \[2\] Reasoning or memorization? unreliable results of reinforcement learning due to data contamination. arXiv 2025\.
>
> \[3\] Benchmarking Benchmark Leakage in Large Language Models. arXiv 2024\.

---

> ### Author Response · Authors · 2025-11-21
> **[Part 3/4] Author Response**
>
> > **Weakness 2:** I think the performance improvement is somewhat limited: for example, in table 3 and 4, training on a new million scale corpus only yields 3 absolute point improvements, while one would expect bigger improvements (e.g., see the dataset scaling study in the openthoughts paper https://arxiv.org/abs/2506.04178)
>
> **(3) MegaScience is not only more effective, but also far more efficient**
>
> we have added experiments on **OpenThoughts** \[1\], **S1k-1.1** \[2\], and **LIMO** \[3\]. To ensure a fair comparison, we strictly **follow their original training settings** and train all models on **Qwen2.5-7B-Base** under the configurations shown below:
>
> **Training Settings:**
>
> | Dataset  | LR  |  Batch Size | Max Length | Epochs |
> | ------ | -----    |  -----     | ---------------    | ----- |
> | OpenThoughts | 5e-6 | 128 | 32768 | 5 |
> | S1k-1.1 | 1e-5 | 16 | 32768 | 5 |
> | LIMO | 1e-5 | 16 | 32768 | 5 |
>
> **Overall Evaluation Results:**
>
> | Dataset  | Avg. Response Length of Training Datasets  | Avg. Response Length of Benchmarks | General Avg.   |  Specific Avg.  | Math Avg.  | All Avg. |
> | ------ |  -----    |   -----     |  --------      | ----- |   ----- |   ----- |
> | OpenThoughts | 6801 |  13391  |  37.04 | 66.50 | **80.55** | 58.99 |
> | S1k-1.1 | 9878  | 12209  | 39.86 | 66.22 |	78.55 |	59.45 |
> | LIMO | 6984 |  12862 | 35.90 | 59.94 | 76.88 | 54.99 |
> |MegaScience | **693**  | **1345**  |  **43.20** |  **66.55** | 79.61 |  **61.01**|
>
> **High-Difficulty Scientific Reasoning (Computation-Heavy) Benchmarks:**
>
> | Dataset  | Scibench | OlympicArena |
> | ------ |  -----    |    -----     |
> | OpenThoughts | 28.12  | 33.04  |
> | S1k-1.1 | 37.19  | 37.51 |
> | LIMO | 40.46 |  35.30 |
> |MegaScience  | **48.75** | **40.23**  |
>
> **Findings**
>
> 1. **MegaScience still achieves the best average performance**, and is slightly lower than OpenThoughts only in the *Math Avg.* category.
>
> 2. **Datasets such as OpenThoughts, S1k-1.1, and LIMO are less competitive on scientific-reasoning computation tasks.**
>     These datasets primarily target mathematical reasoning and commonly evaluate on benchmarks such as GPQA (multiple-choice), which do not fully reflect scientific computational ability.
>     On the **science-domain, computation-intensive benchmarks** — **SciBench** \[4\] and **OlympicArena** \[5\] — MegaScience achieves **substantially higher scores**, highlighting its advantage in rigorous scientific reasoning.
>
> 3. **MegaScience is not only more effective, but also far more efficient.**
>
>    * **Training efficiency:**
>       Its *Avg. Response Length of Training Datasets* is **\~1/10** of OpenThoughts, S1k-1.1, and LIMO.
>
>    * **Inference efficiency:**
>       Its *Avg. Response Length of Benchmarks* is also **\~1/10**, enabling much faster evaluation and deployment.
>
> 4. Importantly, **the performance improvements of OpenThoughts, S1k-1.1, and LIMO largely rely on *test-time scaling*** — i.e., generating very long chain-of-thought responses distilled from large reasoning models. While this can inflate benchmark scores, it **dramatically increases inference cost**. As discussed in **Lines 065–071**, such *superficial knowledge (data) distillation* not only creates substantial efficiency challenges but may also **hinder more principled, efficient, and generalizable knowledge transfer**.
>
> ---
> \[1\] OpenThoughts: Data Recipes for Reasoning Models. arXiv 2025\.
>
> \[2\] s1: Simple test-time scaling. EMNLP 2025\.
>
> \[3\] LIMO: Less is More for Reasoning. COLM 2025\.
>
> \[4\] SciBench: Evaluating College-Level Scientific Problem-Solving Abilities of Large Language Models. ICML 2024\.
>
> \[5\] OlympicArena: Benchmarking Multi-discipline Cognitive Reasoning for Superintelligent AI. NeurIPS 2024\.

---

> ### Author Response · Authors · 2025-11-21
> **[Part 4/4] Author Response**
>
> > **Question 1:** Can you provide detailed stats for the TextbookReasoning dataset – i.e., breakdown of the domains, the average input and output token length.
>
> Thank you very much for the insightful suggestion.
>
> First, regarding the detailed statistics of the TextbookReasoning dataset: the number of textbooks, the number of chunks, the extracted QA pairs, as well as the changes in duplication, filtering, and decontamination, are all reported in Table 15 and Table 16 in the appendix. Due to space limitations, these tables were placed in the supplementary materials, and we apologize if this caused them to be overlooked. We have already referenced them in the main paper, but we will further clarify this in the revised version.
>
> In addition, you pointed out that providing average input and output token lengths would be very helpful. We fully agree, and we have now included the detailed per-domain statistics below:
>
> | Subject | \# Pairs | Average Question Tokens | Average Answer Tokens |
> | ------ |  -----    |    ----  |  --------------- |
> |  Biology | 52,850 | 27.17 | 295.58
> | Chemistry | 32,157 | 67.43 | 371.32 |
> | Computer Science | 17,742 | 79.54 | 393.34 |
> | Economics | 1,296 | 78.10 | 301.34 |
> | Mathematics | 424,714 | 70.83 | 442.95 |
> | Medicine |  81,638 | 34.45  |  286.59  |
> | Physics | 41,443 | 86.30 | 496.54 |
> | **Total** | 651,840 | 63.80 | 409.66 |
>
> We have incorporated this expanded table into the revised version to improve clarity and completeness (Table 17; Line 172, Line 974-984).
>
> Thank you again for your valuable feedback—it has significantly helped us strengthen the presentation of our dataset.
>
> > **Ethics Concerns:** I think there are potential copyright concerns as the authors directly scrape the textbook PDFs and I don’t think they’ve reported reliable measures to address the copyright issues (line 493, the primary copyright filtering is through LLM detected copyright information, which can be unreliable.)
>
> See General Response to Copyright Concern.

---

### Author Response · Authors · 2025-11-21
**[Part 2/2] General Response to Copyright Concern**

Our copyright-related keyword list is as follows:
```
copyright_related_keywords = {
    'all_rights_reserved': [
        r'all\s*rights\s*reserved',
        r'without.*permission.*prohibited',
        r'reserved\s*rights\s*of',
        r'rights\s*reserved\s*by'
    ],
    'no_reproduction': [
        r'no\s*part.*may\s*be\s*reproduced',
        r'reproduction.*strictly\s*prohibited',
        r'may\s*not\s*be\s*copied',
        r'may\s*not\s*be\s*duplicated',
        r'copying\s*without\s*permission',
        r'unauthorized\s*copying'
    ],
    'no_commercial_use': [
        r'commercial\s*use\s*prohibited',
        r'no\s*commercial\s*use',
        r'strictly\s*non-?commercial',
        r'not\s*for\s*sale',
        r'not\s*for\s*commercial\s*distribution'
    ],
    'no_distribution': [
        r'distribution.*prohibited',
        r'no\s*distribution',
        r'may\s*not\s*be\s*shared',
        r'unauthorized\s*distribution'
    ],
    'no_modification': [
        r'modification.*prohibited',
        r'no\s*modification',
        r'alteration.*prohibited',
        r'may\s*not\s*be\s*edited',
        r'may\s*not\s*be\s*modified'
    ],
    'no_derivative_works': [
        r'derivative\s*works.*prohibited',
        r'no\s*derivative\s*works',
        r'adaptations.*prohibited',
        r'transformation.*prohibited'
    ],
    'publisher_exclusive': [
        r'exclusive\s*property\s*of',
        r'exclusively\s*owned\s*by',
        r'property\s*of\s*(the\s*)?publisher',
        r'belongs\s*to\s*(the\s*)?publisher'
    ],
    'written_permission_required': [
        r'written\s*permission.*required',
        r'prior\s*written\s*consent',
        r'obtain\s*permission\s*before\s*(use|reproduction)',
        r'permission\s*in\s*writing\s*from'
    ],
    'copyright_violation_warning': [
        r'copyright\s*violation',
        r'infringement.*legal\s*action',
        r'unauthorized\s*use.*prosecuted',
        r'violation.*of\s*copyright\s*law',
        r'copyright\s*protected\s*material'
    ],
    'educational_only_strict': [
        r'educational\s*use\s*only.*no\s*other',
        r'strictly\s*educational',
        r'for\s*classroom\s*use\s*only',
        r'academic\s*use\s*only'
    ],
    'internal_use_only': [
        r'internal\s*use\s*only',
        r'confidential.*internal',
        r'for\s*internal\s*circulation',
        r'not\s*for\s*public\s*release'
    ],
    'no_ai_training': [
        r'artificial\s*intelligence.*prohibited',
        r'machine\s*learning.*prohibited',
        r'ai\s*training.*prohibited',
        r'may\s*not\s*be\s*used\s*to\s*train\s*(ai|machine\s*learning)',
        r'not\s*authorized\s*for\s*(ai|ml)\s*training'
    ],
    'no_data_mining': [
        r'data\s*mining.*prohibited',
        r'text\s*mining.*prohibited',
        r'scraping.*prohibited',
        r'may\s*not\s*be\s*harvested',
        r'no\s*automated\s*(scraping|analysis|collection)'
    ],
    'drm_protected': [
        r'drm\s*protected',
        r'digital\s*rights\s*management',
        r'copy\s*protection',
        r'encrypted\s*content',
        r'digital\s*watermark'
    ],
    'redistribution_limited': [
        r'not\s*authorized\s*for\s*redistribution',
        r'may\s*not\s*be\s*posted\s*online',
        r'posting\s*on\s*the\s*internet.*prohibited',
        r'not\s*to\s*be\s*shared\s*electronically'
    ],
    'legal_notice': [
        r'legal\s*notice',
        r'governed\s*by\s*copyright\s*law',
        r'subject\s*to\s*copyright\s*restrictions'
    ]
}
```

---

### Author Response · Authors · 2025-11-21
**[Part 1/2] General Response to Copyright Concern**

We sincerely thank all reviewers for their careful reading of our manuscript and for raising important concerns regarding copyright and data sourcing. We greatly appreciate the opportunity to clarify our procedures and provide additional details.

In terms of copyright detection, we have built a dedicated test set along with a rule-based + LLM-based detection pipeline. While we cannot guarantee perfect detection across all data, on our test set, our pipeline successfully identifies **100% of the textbooks with copyright restrictions**, and 10% of unrestricted samples are mistakenly flagged as restricted. This demonstrates the **high recall** and the **strictness** of our detection procedure.

In our **Ethics Statement**, we clearly emphasize the following points:

1. Our dataset is **for research use only**, and when released, it will be distributed under the **CC BY-NC-SA 4.0 license**, which explicitly prohibits commercial use.

2. We will **open a Feedback and Removal Request channel**. After the dataset is released, if we receive any feedback indicating that certain data may violate copyright or related rights, we will promptly remove the corresponding content.

To construct our **test set**, we manually curated **50 textbooks with known copyright restrictions** as positive samples and **50 textbooks without any copyright limitations** as negative samples. This dataset is used to evaluate and validate the accuracy of our detection pipeline. Our pipeline successfully identifies **100% of the textbooks with copyright restrictions**, and 10% of unrestricted samples are mistakenly flagged as restricted. This demonstrates the **high recall** and the **strictness** of our detection procedure.

The detailed **rule + LLM detection procedure** is as follows:

1. We manually construct a **copyright-related keyword list**. Using regular expressions, we scan all documents and flag those containing these keywords as **candidate documents** potentially subject to copyright restrictions.

2. For each book, we also include its **first 5 and last 5 segments** (each being a 4096-token chunk) in the candidate list, since copyright information is typically found at the beginning or end of a book.

3. We then use Llama-3.3-70B-Instruct to determine whether each document is **permitted for public use and redistribution** and **does not prohibit AI training**, outputting either “Yes” (allowed) or “No” (not allowed). If **any document** within a book is classified as “No” the entire book is excluded from our dataset. To improve robustness, each candidate document is evaluated **twice** independently, and only documents labeled “Yes” in both passes are retained.

The LLM detection prompt we use is as follows:
```
You are an expert analyst specializing in copyright, licensing, and terms of use for textual documents. Your task is to determine the permissions granted for a given document.

Task: Analyze the provided document text and determine whether it is explicitly permitted for public use and redistribution **AND** does not explicitly prohibit its use for Artificial Intelligence training.

**Guidelines:**

1.  **Focus on Explicit Statements:** Pay close attention to words and phrases related to:
    *   **Permission:** "public domain," "open access," "free to share," "distribute," "Creative Commons" (e.g., CC BY, CC0), "permitted," "allowed," "you may use this for..."
    *   **Prohibition/Restriction:** "all rights reserved," "prohibited," "may not," "for personal use only," "not for redistribution," "requires permission," "no derivative works," and crucially, any mention of "AI," "machine learning," "ML," "training," "data mining," or "algorithmic use" in a restrictive context.

2.  **Decision Logic:**
    *   Your decision should be **"YES"** only if the document **clearly indicates it is allowed for public use/redistribution** AND **contains no explicit prohibition against AI/ML training**.
    *   If the document is silent on AI training but allows redistribution, default to "YES" unless other restrictive terms contradict it.
    *   Your decision should be **"NO"** if the document is clearly restricted (e.g., all rights reserved, for personal use only) OR if it **explicitly forbids use for AI/ML training**, even if it seems publicly available.

Output Format: You MUST output your response in the following exact format. Do not add any other text before or after.

Analysis:
<Provide a concise, step-by-step explanation of your reasoning based on the document's text. Point to specific phrases or the lack thereof that led to your decision.>

Decision:
<YES/NO>
- YES: The document is permitted for public use and redistribution and does not prohibit AI training.
- NO: The document is NOT permitted for public use and redistribution OR it explicitly prohibits AI training.

Input:
The document:
`<DOCUMENT>`
```

---

### Author Response · Authors · 2025-12-03
**Summary General Response**

We thank all reviewers, the AC, and the PCs for their time and thoughtful feedback. We are grateful for the positive assessments highlighting:

- **Clear research motivation** (Reviewers BVV2, Dmgy)
- **Solid open-source contribution on post-training data** (Reviewers vNaP, BVV2, Dmgy)
- **Extensive ablations for each component of the data curation pipeline** (Reviewers vNaP, BVV2, ZKqy, Dmgy)
- **Comprehensive evaluation across domains** (Reviewer BVV2)
- **Strong and convincing results demonstrating the high quality of the proposed dataset** (Reviewers ZKqy, Dmgy)
- **A well-organized and well-written paper** (Reviewers BVV2, ZKqy, Dmgy)

Below we briefly summarize the main shared concerns and how our revision addresses them:

1. **Copyright and Licensing Concerns**:  As detailed in our “**General Response to Copyright Concern**,” we have constructed a dedicated test set and designed a combined rule-based + LLM-based detection pipeline. While perfect detection for all data cannot be guaranteed, on our test set the pipeline successfully identifies **100%** of textbooks with copyright restrictions, and only **10%** of unrestricted samples are mistakenly flagged as restricted. This suggests high recall and a deliberately conservative (strict) detection procedure. In addition, our dataset is released for **research use only**, and we will provide a Feedback and Removal Request channel to promptly handle any concerns from content owners. (Response to Reviewer vNaP, BVV2, ZKqy)
2. **Comparison with Related Work (e.g., OpenThoughts, S1k-1.1, LIMO)**: We have added experiments on OpenThoughts, S1k-1.1, and LIMO. For a fair comparison, we strictly follow their original training setups and train all models on **Qwen2.5-7B-Base**. The new results show that: (i) Our **MegaScience** dataset still achieves the best average performance; (ii) OpenThoughts, S1k-1.1, and LIMO are less competitive on scientific reasoning and computation benchmarks; (iii) MegaScience is not only more effective, but also significantly more efficient in both training and inference. (Detailed results refer to Response to Reviewers vNaP, BVV2)
3. **Human Evaluation of LLM Refinement**: We conducted a manual evaluation on 100 randomly sampled instances from TextbookReasoning. Each instance includes an original question–answer pair and the corresponding pair refined by DeepSeek-V3. Human annotators performed pairwise comparisons on question quality, answer quality, error rate, and error taxonomy. The results indicate that, while DeepSeek-V3 can occasionally introduce hallucinations, missing information, or undesirable references, the **majority** of refined questions and answers are either strictly better than or at least comparable to the originals. The observed error rate is approximately **5–7%** in our setting. These findings have been incorporated into the revised version (Appendix D, Lines 858–925). (Response to ZKqy, Dmgy)
4. **Novelty and Contribution of the Work**: We would like to highlight that the central contribution of our work lies in its **open-science value**. We provide the first fully open-source, high-quality scientific reasoning dataset that functions as a **reproducible and extensible research infrastructure** for the community. Building such a dataset is resource-intensive and costly; academic groups with limited compute often cannot construct comparable corpora from scratch, while industry-grade datasets are typically not publicly available. Importantly, models trained on our dataset with **SFT alone** can surpass the performance of the official **Qwen3-Instruct** model, demonstrating both the effectiveness and generality of our data.
Beyond openness, MegaScience also directly addresses key limitations of prior scientific reasoning datasets: (i) unreliable benchmark evaluation, (ii) incomplete decontamination, (iii) low-quality reference answers, and (iv) superficial data distillation. In summary, we believe novelty does not solely reside in algorithmic innovation — it can also emerge through open-science infrastructure that fundamentally enables reproducibility, accessibility, and progress in a research field. (Response to Reviewer vNaP)
5. **Magnitude of Performance Improvements**: Qwen3-Instruct is a very strong, heavily optimized, industry-grade baseline, so even modest absolute gains are non-trivial. From this perspective, achieving an **average improvement of ≈3 points** with a fully open and contamination-verified dataset is already meaningful. Our evaluation spans a broad suite of **14** scientific benchmarks, which naturally includes tasks where performance gaps are smaller. At the same time, Table 19 shows that on several **high-difficulty scientific reasoning benchmarks**, our data yields substantial gains: GPQA **+5.56%**, SciBench **+9.69%**, OlympicArena **+5.97%**. (Response to Reviewer vNaP).

We hope these clarifications better convey our contributions and their significance.

---

### Meta-Review · Area_Chair_n71h · 2026-01-05

**Summary:**

The paper introduces "MegaScience" and "TextbookReasoning," a large-scale data curation project designed to enhance the scientific reasoning capabilities of LLMs. The authors construct a pipeline involving the scraping of over 12,000 university-level textbooks, extracting QA pairs, and refining them using strong models (DeepSeek-V3) to create a dataset of 1.25 million instances. The work demonstrates that models trained on this data can outperform official instruction-tuned baselines (e.g., Qwen3-Instruct) on several scientific benchmarks, highlighting the value of high-quality, short-chain-of-thought data.

Despite the authors' solid execution and a strong rebuttal that addressed technical concerns regarding efficiency and data quality, the agreement leans towards rejection due to a critical and unresolved ethical/legal flaw regarding data sourcing. While the resulting resource is empirically effective, the foundation of the work relies on the unauthorized mass crawling of copyrighted textbooks. Reviewers (e.g., vNaP) rightly pointed out that the proposed strategies (LLM-based filtering) are insufficient to guarantee legality. Consequently, accepting this work would set a concerning precedent regarding the use of intellectual property in dataset construction, outweighting its engineering contributions.

**Reviewer Concerns:**

**Concerns Addressed by the Rebuttal:**
•**Efficiency vs. Performance**: Resolved by new experiments comparing against OpenThoughts, demonstrating that MegaScience achieves competitive results on hard tasks (SciBench) while using 1/10th of the token length, proving the value of "Short CoT."
•**Data Quality/Hallucination**: Resolved by a strict human evaluation (100 samples) requested by Reviewers ZKqy and Dmgy, which showed a 95% improvement in answer quality and a low error rate (6%) after refinement.
•**Baselines**: Addressed by clarifying the difficulty of beating the Qwen3-Instruct baseline and highlighting significant gains in specific high-difficulty domains.

**Outstanding Concerns:**
•**Copyright and Data Sourcing (Fatal Flaw)**: This remains the primary reason for rejection. The foundation of this work—mass scraping of 12,000+ university-level textbooks without clear permission—presents a fundamental ethical and legal violation that cannot be later fixed by after-the-fact filtering. Reviewer vNaP correctly noted that LLM-based filtering is legally insufficient and technically unreliable for determining copyright status. Accepting a paper built on such a data acquisition strategy would expose the conference to significant risk and set a problematic precedent for data sourcing standards, regardless of the resulting model's performance.
•**Methodological Novelty:** The contribution is viewed primarily as a resource/engineering effort using standard techniques (PDF extraction, distillation, mixing) rather than a methodological breakthrough. Reviewer vNaP noted that the novelty is limited to the data pipeline itself, which, given the copyright issues, is difficult to support.

**Reviewer Scores:**

I think the scores are 4, 4, 6, 8.
Reviewers vNaP and BVV2 maintained their scores of 4, reflecting that while the rebuttal improved technical clarity, it could not fundamentally resolve the concerns regarding novelty and, more importantly, the ethical implications of the data source. Reviewer Dmgy (6) and ZKqy (8) focused more on the empirical utility, but as AC, I must weigh the ethical risks heavily against the empirical gains.

---

### Decision · Program_Chairs · 2026-01-26

Reject